# Cloning southern corn rust resistant gene *RppK* and its cognate gene *AvrRppK* from *Puccinia polysora*

Gengshen Chen [1,8], Bao Zhang [1,8], Junqiang Ding [2,3,8], Hongze Wang [1,8], Ce Deng [2], Jiali Wang [1], Qianhui Yang [1], Qianyu Pi [1], Ruyang Zhang [4], Haoyu Zhai [1], Junfei Dong [1], Junshi Huang [1], Jiabao Hou [1], Junhua Wu [1], Jiamin Que [1], Fan Zhang [1], Wenqiang Li [1], Haoxuan Min[1], Girma Tabor [5], Bailin Li [5], Xiangguo Liu[6], Jiuran Zhao [4], Jianbing Yan [1,7] & Zhibing Lai [1,7]

Broad-spectrum resistance has great values for crop breeding. However, its mechanisms are largely unknown. Here, we report the cloning of a maize *NLR* gene, *RppK*, for resistance against southern corn rust (SCR) and its cognate *Avr* gene, *AvrRppK*, from *Puccinia polysora* (the causal pathogen of SCR). The *AvrRppK* gene has no sequence variation in all examined isolates. It has high expression level during infection and can suppress pattern-triggered immunity (PTI). Further, the introgression of *RppK* into maize inbred lines and hybrids enhances resistance against multiple isolates of *P. polysora*, thereby increasing yield in the presence of SCR. Together, we show that *RppK* is involved in resistance against multiple *P. polysora* isolates and it can recognize AvrRppK, which is broadly distributed and conserved in *P. polysora* isolates.

Maize (*Zea mays* L) is one of the crops with the highest production and a major source of calories and proteins for livestock and humans worldwide. However, the grain yield and quality of maize are seriously reduced by many diseases around the world. The most efficient way to control these diseases is to develop maize lines carrying resistance genes. The identification and cloning of resistance genes are critical steps toward that goal. However, only sixteen resistance genes (*Hm1*, *Htn1*, *Ht2*, *Ht3*, *Rp1-D21*, *RppC*, *RabGD1α*, *ZmABP1*, *ZmAuxRP1*, *ZmCCoAOMT2*, *ZmCCT*, *ZmFBL41*, *ZmMM1*, *ZmREM1.3*, *ZmTrxh*, *ZmWAK*) have been cloned from maize[1–15], and very few of them have been proven to have breeding value.

Plant innate immunity consists of pattern-triggered immunity (PTI) and effector-triggered immunity (ETI)[16]. PTI is triggered by the recognition of microbial components by cell surface-localized pattern-recognition receptors (PRRs), while ETI is activated by the recognition of microbial effector proteins directly or indirectly by intracellular nucleotide-binding leucine-rich repeat receptors (NLRs)[16]. ETI activates more rapid and stronger immunity than PTI[17,18], although they share similar downstream cellular responses[19,20]. Therefore, the deployment of *NLR* genes in crop cultivars has been the major approach for improving disease resistance in crop breeding[21]. Unfortunately, the resistance conferred by *NLR* genes is often not durable in the field because of changes in pathogen races. Pathogen races without corresponding effector genes (known as avirulence, *Avr*, genes) accumulate quickly, and pathogen races may evolve through the mutation or deletion of the corresponding *Avr* gene, resulting in the breakdown of resistance conferred by *NLR* genes[16,21,22]. For example, the Ug99 race group of *Puccinia graminis* f. sp. *tritici* has successfully broken through the resistance of wheat cultivars conferred by the *Sr31*, *Sr24*, *Sr36*, and *SrTmp* resistance genes[23–26]. To achieve broad-spectrum

[1]National Key Laboratory of Crop Genetic Improvement, Huazhong Agricultural University, 430070 Wuhan, Hubei, China. [2]College of Agronomy, Henan Agricultural University, 450002 Zhengzhou, Henan, China. [3]The Shennong Laboratory, 450002 Zhengzhou, Henan, China. [4]Maize Research Center, Beijing Academy of Agriculture and Forestry Sciences (BAAFS), 100097 Beijing, China. [5]Corteva Agriscience, Johnston, IA 50131, USA. [6]Institute of Agricultural Biotechnology, Jilin Academy of Agricultural Sciences, 130033 Changchun, Jilin, China. [7]Hubei Hongshan Laboratory, 430070 Wuhan, Hubei, China. [8]These authors contributed equally: Gengshen Chen, Bao Zhang, Junqiang Ding, Hongze Wang. ✉e-mail: yjianbing@mail.hzau.edu.cn; zhibing@mail.hzau.edu.cn

resistance, the pyramiding of multiple *NLR* genes has been applied in breeding[22]. For example, the pyramiding of yellow rust resistance genes (*Yr5* and *Yr15* or *Yr64* and *Yr15*) in wheat resulted in broad-spectrum resistance against all tested stripe rust races[27,28]. Another way to obtain broad-spectrum resistance is to deploy *NLR* genes recognizing core effectors, which are widely distributed in most races of a particular pathogen and are required for their virulence in crops[21]. Although much work has been performed to achieve that goal[29,30], no core effector gene-*NLR* gene pairs have been identified in major pathogens and crops. Therefore, no direct evidence has been presented to prove that this strategy is practicable.

Owing to changes in climate and cropping practices, southern corn rust (SCR) caused by *Puccinia polysora* UnderW. has become a major maize disease in the USA, Canada, Brazil, and China[31–35]. Under favorable conditions, SCR can cause yield losses of more than 50%, which seriously threatens maize production and food security[31,32,35–38]. Similar to other rust fungi, *P. polysora* behaves as an obligate biotroph that only extracts nutrients from living cells. Although eleven maize dominant resistance genes (*Rpp1* to *Rpp11*) and eight maize major resistance QTLs (*RppC*, *RppCML470*, *RppD*, *RppM*, *RppP25*, *RppQ*, *RppS*, and *RppS313*) against *P. polysora* have been reported, only *RppC* was cloned[14,31,33,37,39–44].

In this work, we clone a maize *CC-NB-LRR* gene, *RppK*, involved in resistance against *P. polysora* by map-based cloning, and its cognate *Avr* gene, *AvrRppK*, from *P. polysora*. *AvrRppK* is highly conserved in all tested *P. polysora* isolates and can suppress chitin-induced PTI. Furthermore, the *RppK* gene has great value for breeding because maize inbred lines or hybrids introgressed with the *RppK* gene exhibit stronger resistance against *P. polysora* and higher yields than those without the *RppK* gene in the presence of *P. polysora*.

## Results

### Map-based cloning of *RppK*

K22 is a maize inbred line with durable and complete resistance against SCR in China over the last 30 years (Fig. 1a and Supplementary Fig. 1a). To study the genetic mechanism underlying its resistance against SCR, K22 was crossed with the susceptible inbred line DAN340 to generate a K22 × DAN340 $F_{6:7}$ population[45,46], and the disease phenotypes of these plants were evaluated, based on the SCR disease scale (Supplementary Fig. 1b, c). QTL analysis identified one major QTL, *RppK*, on the short arm of chromosome 10 that accounted for 68% of the phenotypic variation in resistance against SCR (Fig. 1b and Supplementary Table 1). To fine-map *RppK*, we genotyped and evaluated the SCR resistance of 402 recombinant lines isolated from 3392 HIF (heterogeneous inbred family)-derived $F_2$ populations[47]. *RppK* was mapped to an interval of ~18.3 kb delimited by the markers SNP20 and SNP5 based on the B73 RefGen_v4 genomic sequence (Fig. 1c, d and Supplementary Fig. 2).

The *RppK*-linked molecular markers RUST7-5 and RUST9-4 were then used to screen BAC libraries of K22 and DAN340. The sequencing and annotation of five BAC clones from K22 revealed that the candidate region between markers SNP20 and SNP5 in K22 spanned 29.8 kb and contained three genes (*R1*, *R2*, and *R3*) (Supplementary Fig. 3). The three candidate genes belong to the *CC-NLR* gene family with more than 96% nucleotide identity with each other (Supplementary Figs. 4 and 5). In contrast, the candidate region in the susceptible inbred line DAN340 was 20.2 kb and harbors only one *CC-NLR* gene; and, this gene was named as *DAN340 R gene homologous to R3*, or *DR3* (Supplementary Figs. 4 and 5). Based on the DNA sequences of the candidate regions in K22 and DAN340, we developed more markers to genotype recombinant plants. The candidate region for the *RppK* region was ultimately delimited between markers RRD103 and RRD111, which encompassed only the *R2* and *R3* genes (Fig. 1d, e). We subsequently sequenced the *DR3* and *R3* genes in 21 recombinant plants and found that 11 of them showed recombination between *DR3* and the *R1*, *R2* or

*R3* gene (Supplementary Fig. 6). Among the 11 recombinant lines, five lines (4H1074, 4H1505, 4H1083, 4H1028 and 4H1213) carrying the *R3* gene showed resistance to SCR, whereas the other six lines, containing either no *R3* gene (5H2661, 4H1285 and 4H1169) or only a partial *R3* gene (4H1652, 4H1474 and 4H943), exhibited a susceptible response to SCR (Supplementary Fig. 6). These results indicated that *R3* gene is responsible for *RppK* resistant against SCR. To determine whether *R2* or *R3* contributed to SCR resistance, we cloned genomic DNA fragments of 12.6 and 11.4 kb for *R2* and *R3*, respectively, from K22; these fragments contained whole genes with their native promoter and terminator sequences (Fig. 1f and Supplementary Fig. 7a). The two fragments were then transformed into the susceptible inbred line KN5585. Two independent $T_1$ families of *R2* or *R3* gene transgenic plants were tested for resistance to natural infection by *P. polysora* in the field in Hainan over three years (2017, 2019 and 2020). *R2* transgenic plants showed similar SCR susceptibility to the nontransgenic plants (Supplementary Fig. 7b, c), whereas all *R3* transgenic plants from two independent families showed stronger resistance to SCR than the nontransgenic plants (Fig. 1g, h and Supplementary Fig. 8). In order to confirm the resistance function of *R3* gene, *R3* transgenic plants and nontransgenic plants were inoculated with five different isolates of *P. polysora* in growth room (Supplementary Fig. 9a). Moreover, *R3* transgenic plants were more resistant to all five *P. polysora* isolates than the nontransgenic control plants (Supplementary Fig. 9b–f). We conclude that *R3* is the QTL encoding resistance gene against SCR, and was renamed as *RppK*.

### *RppK* presents great values for maize breeding

To explore natural *RppK* variation in maize accessions, we used four *RppK*-related molecular markers (R8.65, R8.63, Del13K and R8.61) (Supplementary Fig. 10a, b) to screen ~500 diverse maize inbred lines from an association mapping panel[48] and found that only 17 of them contained the *RppK* gene (Supplementary Fig. 10, Supplementary Table 2 and Supplementary Data 1). Also, we screened 168 teosinte lines and 288 landrace lines using the functional marker R8.63, which did not amplify related sequences (Supplementary Data 2 and 3). Further, we checked 74 commercial maize hybrid lines and found that only five of them contained the *RppK* gene (Supplementary Fig. 11 and Supplementary Data 4). The low frequency of the *RppK* gene in the current maize collections indicates that the allele has not broadly spread in maize lines, which highlights the potential value of the *RppK* gene in improving SCR resistance in maize breeding. Therefore, we introgressed the *RppK* allele into ten elite inbred lines via backcrossing and evaluated their SCR resistance in Hainan, China. The plants harboring the *RppK* gene were significantly more resistant to SCR than the plants without *RppK* (Supplementary Fig. 12). To evaluate the breeding value of *RppK* in hybrids, we performed repeated backcrossing and molecular marker-assisted selection to introduce the *RppK* gene into the two parental lines of the maize hybrid JK968 (Jing724 × Jing92), which has been planted seven million hectares in China over the past decade[49]. In the field, the JK968 hybrid lines carrying *RppK* were more resistant to SCR than the original JK968 hybrid (Fig. 2a–d). As a result, the grain yields of the JK968 lines carrying *RppK* were 11.9% (with one *RppK* allele, JK968a) and 17.1% (with two *RppK* alleles, JK968b) higher than that of the original JK968 hybrid without *RppK* in the presence of SCR without significant changes in major agronomic traits (Fig. 2c–f and Supplementary Fig. 13). We also improved the SCR resistance of four other hybrid lines with different genetic backgrounds by using the same strategy and got similar results (Supplementary Figs. 14 and 15). Moreover, we generated four additional hybrids by crossing transgenic $KN5585^{RppK}$ lines with four maize inbred lines (B73, Mo17, IL1, and IL2). Consistent with the previous results, the *RppK* gene enhanced maize resistance against SCR and increased the grain yield by 4.51% to 15.6% under different SCR disease conditions with no yield penalty in the absence of SCR

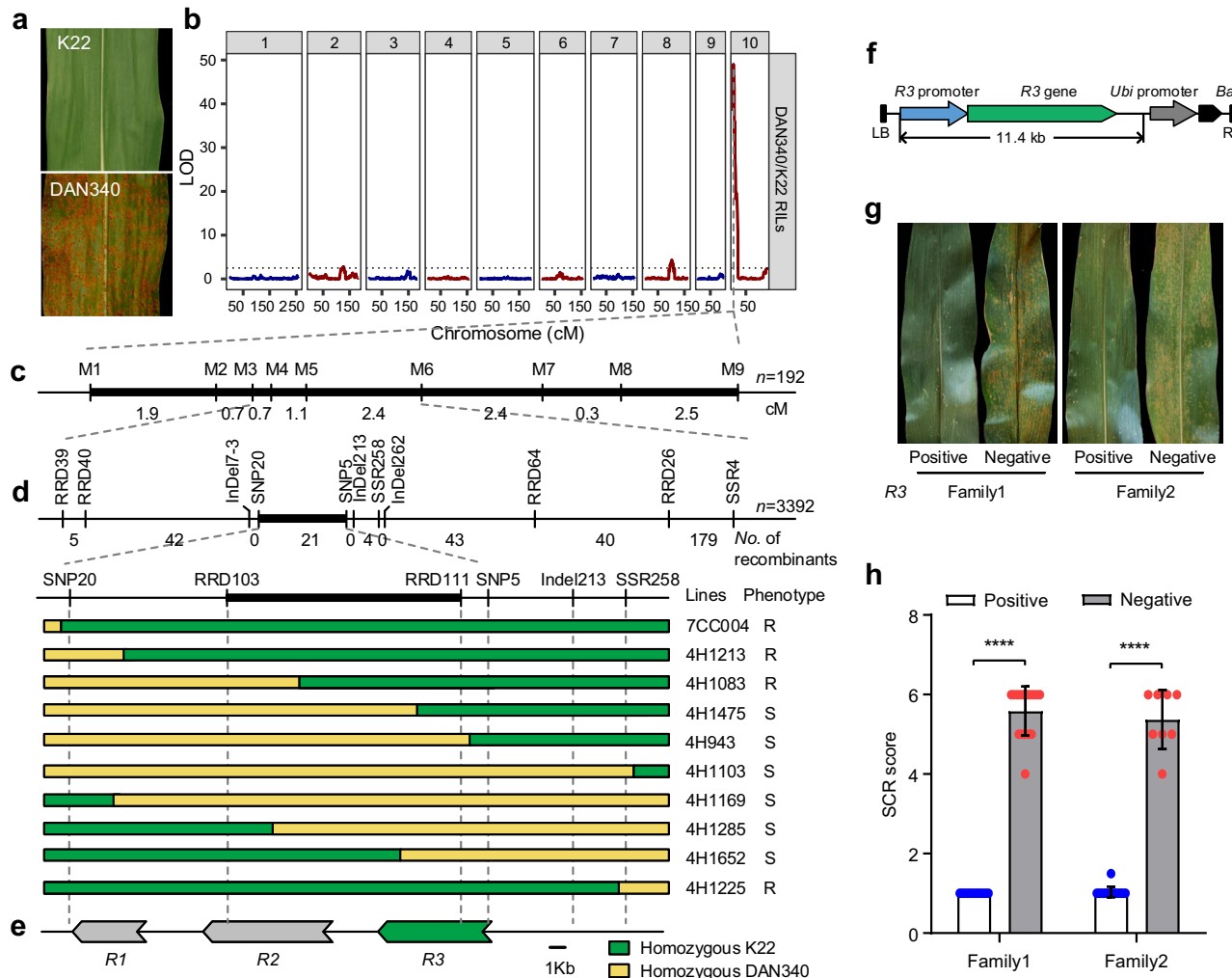

**Fig. 1 | Map-based cloning of the *RppK* gene, conferring resistance against southern corn rust. a** Southern corn rust (SCR) disease phenotypes of maize inbred lines K22 and DAN340. **b** SCR-related *QTL* locations in the genetic population of DAN340/K22 RILs. LOD, logarithm of odds. **c** The linkage mapping of the *RppK* locus based on 192 families delimited it to the region between markers M3 and M6. The numbers below the black rectangle indicate the relative genetic distance (cM) between markers. **d** High-resolution linkage analysis performed by genotyping 3392 plants of a heterogeneous inbred family (HIF) delimited *RppK* to a 15.2 kb interval flanked by markers RRD103 and RRD111. The number of recombinants is shown below the markers. Green segments represent the K22 allele, and yellow segments represent the DAN340 allele. The names of lines and the corresponding phenotypes are provided next to the haplotype. R, resistant; S, susceptible. **e** The *RppK* region contains three predicted *NLR* genes, which encode CC-NB-LRR proteins. Scale bar, 1 kb. **f** Structure of the *R3* genomic sequence construct used for generating transgenic maize plants. LB, left border; RB, right border; Bar: Bialaphos Resistance gene. The construct contains the entire *R3* genomic DNA sequence, including its 3.1 kb promoter region and a 2.4 kb downstream region. The back arrow indicates the Bar gene. **g, h** The disease phenotypes and scores of two independent transgenic plant families carrying the *R3* genomic sequence. Values are means ± SDs; *n* = 15, 17, 14, and 8 individual plants of positive and negative lines of family 1 and family 2, respectively. ****P* < 0.0001 (Student's *t*-test, two-tailed; *P* = 1.2596E-15, 5.4354E-07). Source data are provided as a Source Data file.

(Supplementary Figs. 14–16). These results indicate the high breeding value of *RppK* in maize.

### The *RppS* gene is an allele of *RppK*

There are eight other SCR resistance *QTLs* mapped to the same region as *RppK* on the short arm of chromosome 10[33,37,40–44]. We obtained resistant donor lines for five of those *QTLs* (CML470, CML496, P25, QI319, and SCML205) and analyzed their genetic relationships with *RppK*. By genotyping these lines using four markers (Del13K, R8.65, R8.63, and R8.61), we discovered that only SCML205, which carried the *RppS* locus, showed the same genotype as K22 (Supplementary Table 3). Furthermore, sequencing analysis revealed that SCML205 carried a *RppK* genomic DNA sequence almost identical to the *RppK* gene from K22, with only a 2 bp-indel difference in the second intron region between the two alleles (Supplementary Fig. 17). Therefore, the *RppS* locus from SCML205 is likely to be identical to *RppK* in K22.

### Cloning of AvrRppK from *P. polysora*

In order to identify *AvrRppK*, we used PacBio sequencing to obtain the full-length mRNA sequences of *P. polysora* isolated from germinated urediospores. Through extensive bioinformatic analysis[50,51], we identified 965 *P. polysora* genes (*PPGs*) predicted to encode secretory proteins, and successfully cloned 338 of them with a cysteine content greater than 1.8% (Supplementary Fig. 18).

The recognition of an Avr protein by the corresponding R protein typically activates the hypersensitive response (HR), which is a rapid cell-death phenotype. A powerful tool for *Avr* gene identification has been developed based on transient protoplast expression of the *LUC* reporter gene, whose expression level is used as an indicator of cell viability[52,53]. To identify the interactor of the *RppK* gene, we coexpressed the 338 cloned *PPGΔSPs* (*PPG* genes without signal peptide region) with the *LUC* gene in the protoplasts of transgenic *RppK* plants and nontransgenic control plants. *Rp1-D21*, an HR-activating *NLR*

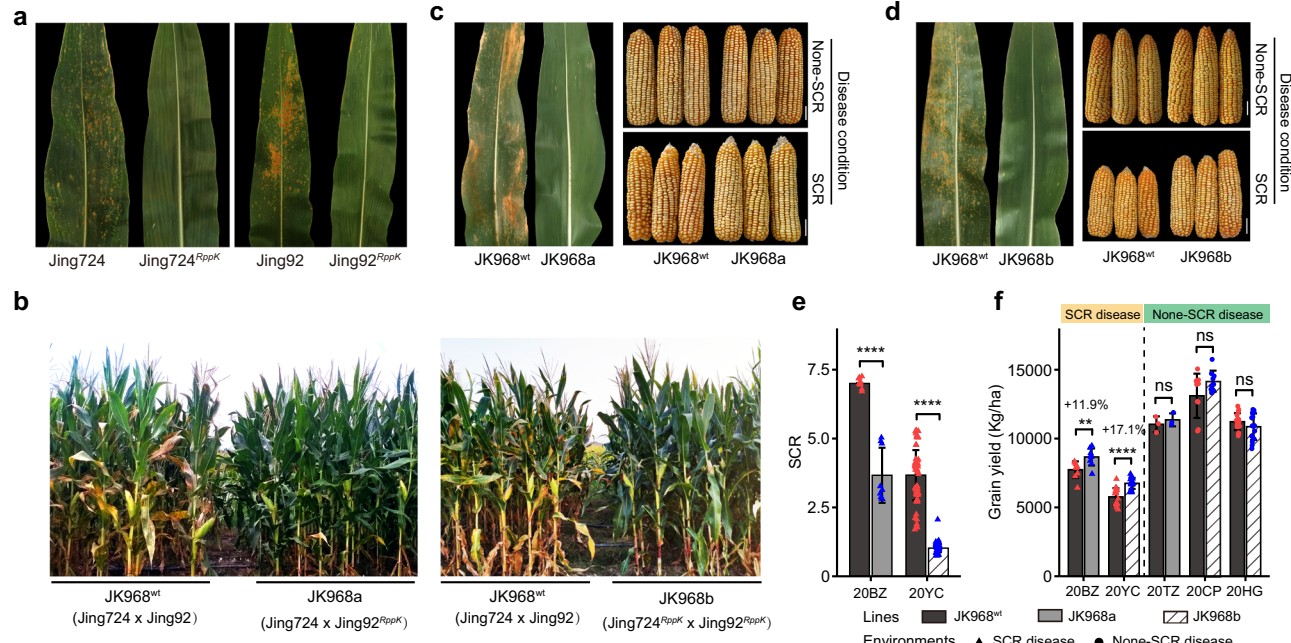

**Fig. 2 | *RppK* improves the grain yield of maize hybrid JK968 (Jing724 × Jing92) in the presence of SCR disease. a** SCR disease phenotypes of wild-type lines (Jing724 and Jing92) and improved lines (Jing724*RppK* and Jing92*RppK*) carrying the *RppK* allele under field conditions. **b** The field performance of hybrid JK968*wt*, improved hybrid JK968a (Jing724 × Jing92*RppK*) and improved hybrid JK968b (Jing724*RppK* × Jing92*RppK*) in the field in the presence of SCR disease. Hybrid JK968a was derived from a cross between Jing724 and Jing92*RppK* and Hybrid JK968b was derived from a cross between Jing724*RppK* and Jing92*RppK*. **c** SCR disease phenotypes of hybrid JK968*wt* and hybrid JK968a under field conditions and ear phenotypes of hybrid JK968*wt* and hybrids JK968a in the presence or absence of SCR disease. Scale bars = 3 cm. **d** SCR disease phenotypes of hybrid JK968*wt* and hybrid JK968b under field conditions and ear phenotypes of hybrid JK968*wt* and hybrids JK968b in the presence or absence of SCR disease. Scale bars = 3 cm. **e** The SCR disease

phenotypes of hybrids JK968*wt*, JK968a, and JK968b in field trials were evaluated in Bozhou (20BZ, $P = 8.48818E-06$) and Yongcheng (20YC, $P = 1.38435E-33$) China, in 2020. **f** The grain yields of JK968*wt*, JK968a, and JK968b in the presence or absence of SCR disease in field trials in China in 2020. SCR disease conditions: 20BZ ($n = 9$ repeat plots; 38 plants of each line in each plot; $P = 0.005498515$), Bozhou, China, in 2020; 20YC ($n = 16$ repeat plots; 11 plants of each line in each plot; $P = 1.96453E-05$), Yongcheng, China, in 2020; non-SCR disease conditions: 20TZ ($n = 3$ repeat plots; 38 plants of each line in each plot; $P = 0.4957928$), Tongzhou, China, in 2020; 20CP ($n = 9$ repeat plots; 38 plants of each line in each plot; $P = 0.100474299$), Changping, China, in 2020; 20HG ($n = 16$ repeat plots; 11 plants of each line in each plot; $P = 0.212607571$), Huanggang, China, in 2020. Values are means ± SDs. **$P < 0.01$; ****$P < 0.0001$ (Student's *t*-test, two-tailed). Source data are provided as a Source Data file.

gene[54], was used as a positive control (Fig. 3a). Only *PPG1259ΔSP* induced a strong HR in the protoplasts of transgenic *RppK* plants and no HR in those of nontransgenic control plants (Fig. 3b and Supplementary Fig. 19). Further, we deployed the transient expression system in *Nicotiana benthamiana*. The *RppK* genomic DNA clone was coexpressed with the *PPG1259ΔSP* gene in *N. benthamiana*. We observed a clear HR when the *RppK* and *PPG1259ΔSP* clones were coinfiltrated (Fig. 3c). In contrast, no HR was triggered by the infiltration of *RppK* genomic DNA or *PPG1259ΔSP* with an empty vector (EV) (Fig. 3c and Supplementary Fig. 20). This indicated that *RppK* induced the HR in a *PPG1259*-dependent manner. Moreover, we generated recombinant constructs encoding GST-PPG1259ΔSP and GST-PPG348ΔSP for the protein infiltration assay. *PPG348ΔSP* was used as a negative control because it did not induce cell death in maize protoplasts with or without *RppK*. These two constructs were expressed in *Escherichia coli*, and the purified recombinant proteins were digested with 3 C PPase to cleave the GST tag (Supplementary Fig. 21). The purified PPG1259ΔSP and PPG348ΔSP proteins were then infiltrated into the leaves of transgenic *RppK* plants and nontransgenic control plants. One day after infiltration, a clear HR was observed in the PPG1259ΔSP-infiltrated leaves of transgenic *RppK* plants but not in the PPG1259ΔSP-infiltrated leaves of nontransgenic control plants (Fig. 3d). As expected, no HR was observed in the PPG348ΔSP-infiltrated leaves of either transgenic *RppK* plants or nontransgenic control plants (Fig. 3d). So, we conclude that the interaction between *PPG1259* encoded AvrRppK protein and RppK could trigger ETI.

We also coexpressed the *RppS* genomic DNA clone with the *AvrRppKΔSP* gene in *N. benthamiana*. A clear HR was observed when

the *RppS* and *AvrRppKΔSP* clones were coinfiltrated and no HR was triggered in the controls (Supplementary Fig. 22). It indicates that *RppS* might recognize *AvrRppK* to trigger ETI.

### AvrRppK of *P. polysora* suppresses plant resistance
*AvrRppK* encodes a 96-aa protein with a predicted secretion signal peptide at the N-terminus but lacks any other known structural domain. BLASTP and BLASTN analysis identified no proteins or genes with similarity to AvrRppK. Therefore, *AvrRppK* might only exist in *P. polysora*. By using single-cell sequencing technology[55], we sequenced and annotated part of the genome of a *P. polysora* strain isolated from Wuhan and identified one contig (8.2 kb) containing the *AvrRppK* gene, which was flanked by a large number of repeat sequences (Supplementary Fig. 23).

To evaluate the variation in *AvrRppK* in natural *P. polysora* isolates, we designed gene-specific molecular markers to examine the genotypes of more than 100 *P. polysora* isolates collected from corn fields in Hainan, Guangxi and Hubei provinces in China. *AvrRppK* gene sequences with the same size were successfully amplified from the isolates. Furthermore, we sequenced the *AvrRppK* coding sequences amplified from 20 isolates from Hainan, 11 isolates from Guangxi, 6 isolates from Hubei and mixed *P. polysora* spores from the field. Interestingly, their coding sequences of *AvrRppK* from all isolates were identical to the *AvrRppK* coding DNA sequence of the Wuhan isolate (Fig. 4a). Therefore, the *AvrRppK* gene was highly conserved in *P. polysora*.

RT-qPCR assays showed that *AvrRppK* was expressed at high levels relative to the *P. polysora Actin* gene at 1, 3, 5, 7, and 9 days after

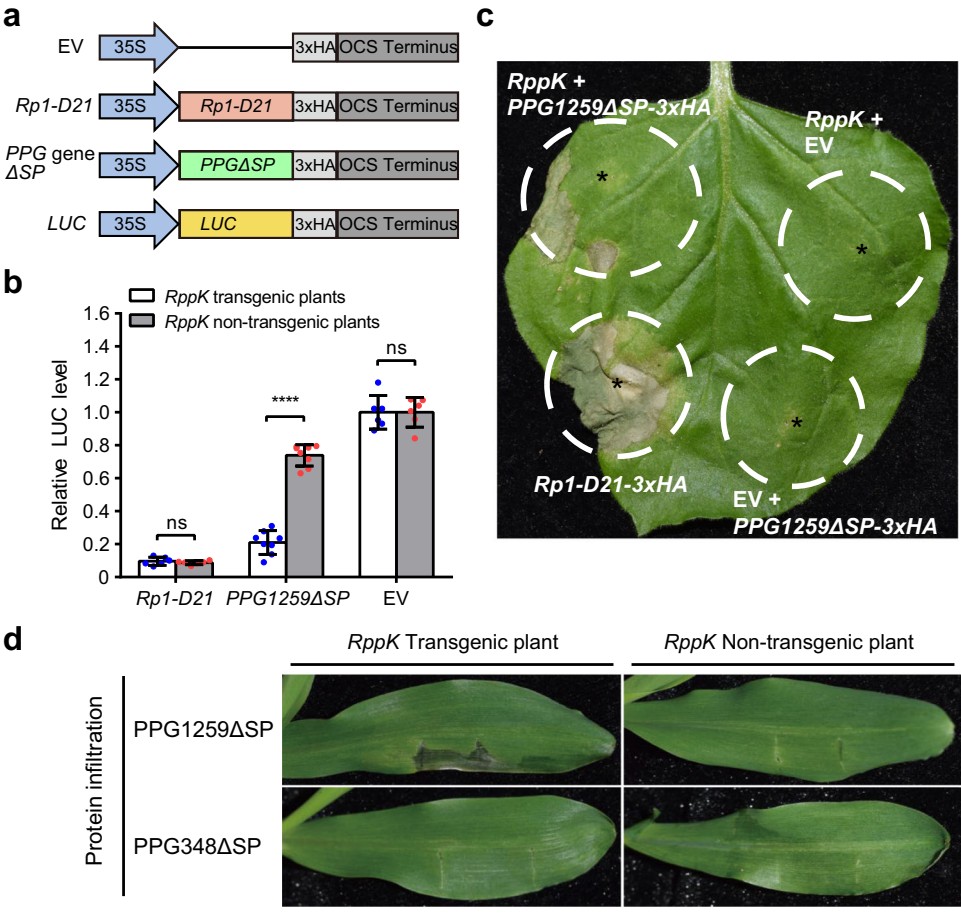

**Fig. 3 | *PPG1259* is the cognate avirulence gene of *RppK*. a** Schematic diagrams of constructs for *AvrRppK* gene screening in transient protoplast expression assays. EV is the empty vector; *Rp1-D21* was used as the positive control; *PPG* genes from *P. polysora* were cloned by using mRNA isolated from *P. polysora*-infected leaves; Δ*SP* indicates deletion of the signal peptide region; and the *LUC* construct was used to evaluate cell viability in the transient protoplast expression assay. **b** *PPG1259*Δ*SP* expression in the protoplasts of *RppK* transgenic plants induced an HR but its expression in the protoplasts of nontransgenic plants did not. *PPG1259*Δ*SP* was coexpressed with the *35s:LUC* construct in the protoplasts of *RppK* transgenic plants or nontransgenic plants; the *Rp1-D21* gene or empty vector (EV) was coexpressed with the *35s:LUC* construct in this assay as a positive or negative control, respectively. Relative LUC levels were measured to indicate the viability of protoplasts. Values are means ± SDs (for expression of *PPG1259*Δ*SP*, *n* = 8; for the others, *n* = 6; *P* = 0.4466, 3.4E-10, 1). **\*\****P* < 0.01; ns, not significant (two-tailed Student's *t*-test). **c** The coinfiltration of the construct carrying the *RppK* genomic DNA sequence with the *35S:PPG1259*Δ*SP* construct induced an HR in *N. benthamiana*. The coinfiltration of the construct carrying the *RppK* genomic DNA construct or the *35S:PPG1259*Δ*SP* construct with the empty vector were performed as negative controls and the infiltration of *Rp1-D21-3×HA* was taken as the positive control. The infiltration sites were labeled with "*". **d** Infiltration of the PPG1259Δ*SP* protein into *RppK* transgenic plants and nontransgenic plants. GST-PPG1259Δ*SP* and GST-PPG348Δ*SP* proteins were expressed in *E. coli* and purified. After digestion with the 3C PPase enzyme to cleave the GST tag, PPG1259Δ*SP* and PPG348Δ*SP* were infiltrated into the leaves of 3-leaf stage *RppK* transgenic plants and nontransgenic plants, respectively. In this assay, *PPG348*Δ*SP*, which did not induce a cell death phenotype in protoplasts of *RppK* transgenic plants or nontransgenic plants, was used as the negative control. Source data are provided as a Source Data file.

inoculation (Fig. 4b). To determine whether *AvrRppK* can enhance disease development, we generated transgenic maize plants overexpressing *AvrRppK*Δ*SP* driven by *ZmUbi* promoter and RT-qPCR results confirmed the expression of *AvrRppK*Δ*SP* in two independent lines (Fig. 4c). After challenged with *P. polysora*, *AvrRppK*Δ*SP* transgenic-positive plants in two independent lines showed more susceptible response to SCR than transgenic-negative plants (Fig. 4d, e) and more *P. polysora* biomass was accumulated in *AvrRppK*Δ*SP* transgenic-positive plants than in transgenic-negative plants (Fig. 4f). It indicates that expression of *AvrRppK* can enhance the development of SCR.

In order to test whether *AvrRppK* can suppress ETI, we coexpressed *AvrRppK*Δ*SP* with *Rp1-D21* in *N. benthamiana*, and clear HR was observed, similar as overexpression of *Rp1-D21* alone (Supplementary Fig. 24), which indicates AvrRppK cannot suppress *Rp1-D21*-mediated ETI. Further, we tried to test whether AvrRppK could suppress PTI. Transgenic plants overexpressing *AvrRppK*Δ*SP* were treated with chitin and two typical PTI responses (MAP kinase signaling and ROS

accumulation) were examined. As the results shown, the chitin-triggered MAP activation was suppressed in *AvrRppK*Δ*SP* transgenic-positive plants, compared with that in transgenic-negative plants (Fig. 4g and Supplementary Fig. 25a). Also, the chitin-triggered ROS accumulation in *AvrRppK*Δ*SP* transgenic-positive plants was weaker than that in transgenic-negative plants (Fig. 4h and Supplementary Fig. 25b). All of these results indicate that *AvrRppK* can significantly suppress chitin-triggered PTI.

## Discussion

The ideal disease resistance in crops would be durable, broad-spectrum resistance, and with no fitness tradeoffs. Here, we reported the cloning of a maize *NLR* gene, *RppK*, for resistance against southern corn rust (SCR), and its cognate *Avr* gene, *AvrRppK*, from *P. polysora* (the causal pathogen of SCR). The introgression of the *RppK* gene into different maize inbred lines and hybrid lines significantly enhanced resistance against multiple *P. polysora* isolates and increased yield in the present of *P. polysora*.

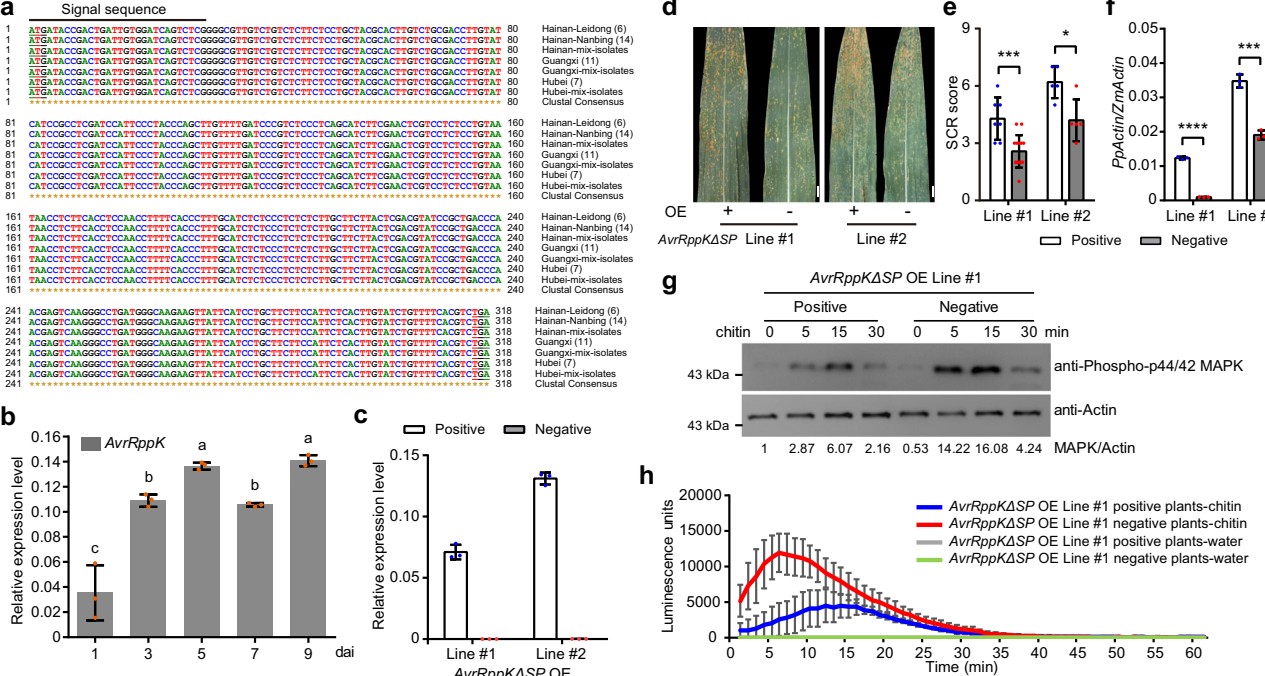

**Fig. 4 | *AvrRppK* suppresses chitin-triggered immunity. a** The *AvrRppK* gene is conserved in all tested *P. polysora* field isolates. DNA samples were isolated from twenty *P. polysora* isolates from Hainan, eleven isolates from Guangxi, six isolates from Hubei and mixed spores collected from Hainan, Guangxi and Hubei, respectively; **b** The expression level of *AvrRppK* during infection. Five-leaf stage B73 plants were inoculated with conidia of the *P. polysora* Wuhan strain. Leaf samples taken at 1, 3, 5, 7, and 9 days after inoculation (dai) were used for RT-qPCR to evaluate the expression levels of *AvrRppK* relative to those of the *P. polysora Actin* gene. Values are means ± SDs ($n = 3$ repeats). One-way ANOVA with Fisher's LSD test (mean ± SD; $n = 3$, biologically independent samples). Different letters indicate significant differences at $P < 0.05$. **c** The transcription levels of *AvrRppKΔSP* measured by RT-qPCR in two independent transgenic maize lines overexpressing *AvrRppKΔSP* ($n = 3$). **d** The SCR disease phenotypes of two independent transgenic maize lines overexpressing *AvrRppKΔSP*. "+" means transgenic-positive plants, "-" means transgenic-

negative plants. **e** The SCR disease scores of two independent transgenic maize lines overexpressing *AvrRppKΔSP* ($n = 7, 14, 5, 5$). **f** The fungal biomass accumulation in two independent transgenic maize lines overexpressing *AvrRppKΔSP* was measured by RT-qPCR amplification of the *P. polysora Actin* gene relative to the *Zea mays Actin* gene ($n = 3$). **g** The chitin-triggered MAP kinase activity was suppressed by *AvrRppK*. Western blot was used to detect MAP kinase activity by anti-Phospho-p44/42 MAPK antibody (dilution 1:1000). The loading control was maize Actin proteins measured by western blot using anti-Actin antibody (dilution 1:5000). Relative MAP kinase activity was normalized to the level of Actin protein. **h** Chitin-induced ROS accumulation was suppressed by *AvrRppK*. The absolute luminescence was used to represent ROS accumulation (for each time point, $n = 6$). Water treatment was taken as the control. For **c**–**h**, values are means ± SDs; *$P < 0.05$, ***$P < 0.001$, ****$P < 0.0001$ (two-tailed Student's *t*-test). These experiments were repeated three times with the similar results. Source data are provided as a Source Data file.

Effector proteins are major virulence factors of biotrophic and hemibiotrophic pathogens that suppress plant immunity[16,56]. In resistant plants, NLR proteins detect effector proteins and trigger ETI[57]. However, effector proteins are subject to rapid evolution through deletions or mutations[21,58]. Consequently, resistant plants may regain susceptibility to pathogens. For example, the disruption of the *AvrSr35* gene by a MITE insertion in *Puccinia graminis* f. sp. *tritici* prevents recognition by the wheat *NLR* gene *Sr35*[59]. However, the core effectors of pathogens are highly conserved and widely distributed in particular pathogens, and the mutation of a core effector significantly suppresses pathogen virulence[58]. Three core effectors in *Ustilago maydis*, Cce1, Pep1 and Vp1, are highly conserved in smut fungi and are required for their virulence[60,61]. In the present study, the coding sequences of *AvrRppK* were found to be identical in all tested *P. polysora* isolates (Fig. 4a), and *AvrRppK* was highly expressed during infection (Fig. 4b). Overexpression of *AvrRppK* in maize significantly suppressed maize resistance against SCR and reduced chitin-mediated PTI (Fig. 4d–h and Supplementary Fig. 25). All of this evidence indicated that AvrRppK might function as an effector. However, it should be noted that we didn't provide genetic evidences to support its role as a core effector.

Since core effectors are highly conserved in specific pathogens, it has been proposed that the deployment of *NLR* genes corresponding to core effectors is a promising strategy for achieving broad-spectrum resistance[21]. Although many core effectors have been identified by comparing the genomic DNA sequences of different strains of a

specific pathogen[58,62,63], no *NLR* genes recognizing these core effectors have been identified. Here, we cloned a maize *NLR* gene, *RppK*. The introgression of *RppK* into multiple elite maize inbred lines and hybrids significantly enhanced resistance against *P. polysora* and increased yields in the present of *P. polysora* (Fig. 2 and Supplementary Figs. 12–15). The improved maize lines were planted in different locations and were challenged with different *P. polysora* isolates, and the results showed that *RppK* functioned well in all tested locations (Supplementary Figs. 12 and 14–16). These results indicate that the deployment of a single *NLR* gene is sufficient to activate resistance against multiple *P. polysora* isolates.

Although the introgression of resistance genes into crops normally causes fitness tradeoffs[64], no yield loss in the absence of *P. polysora* was observed in any of the tested maize hybrid lines after the introgression of the *RppK* gene (Fig. 2c, d, f and Supplementary Figs. 13–15). Furthermore, several important agricultural traits were unaffected by the introgression of the *RppK* gene (Supplementary Figs. 13 and 14). All of these results confirmed that the *RppK* gene presents great values in maize breeding.

In the 500 inbred lines, only seventeen inbred lines contains *RppK* gene (Supplementary Data 1). Six of them belong to non-stiff stalk (NSS) group; eight of them belong to tropical-subtropical (TST) subgroup; and three of them belong to mixed subgroup (Supplementary Table 4)[48]. Also, eleven of the seventeen lines were originated from China, and six of them were originated from CIMMYT, Mexico

(Supplementary Table 4)[48]. Further, only five of 74 commercial maize hybrids in China contain *RppK* gene (Supplementary Fig. 11 and Supplementary Data 4). Based on the information on https://chinaseed114.com, they were released to the market during the last 12 years. So, *RppK* gene is a rare allele and it has not been widely spread in maize inbred lines, which partially explain why *RppK* gene has been durable for more than 30 years in China.

Effectors as the major virulence weapons of pathogens and NLR proteins as the major components of the plant immune system are subjected to evolutionary pressure[65], which consequently results in high variation in effectors and NLR proteins[16,21]. Core effectors are widely distributed in a particular pathogen[21,58]. It indicates that pathogens cannot afford to lose them[58]. To evade the recognition by host resistance proteins, some core effectors show polymorphism in different isolates. Multiple *Avramr1* homologs were identified from different *P. infestans* isolate and they are recognized by different *Rpi-amr1* alleles[66]. In order to explain why core effectors are not deleted in the arm-race, more studies on NLR proteins and core effectors should be conducted in future.

## Methods

### Plant materials
The resistant inbred line K22, derived from K11 × Ye478 cross, is an elite subtropical line and has been used for breeding program in China for ~30 years[67]. The susceptible donor DAN340, with good agronomic traits, is one of the main inbred lines from the Luda Red Cob group in China.

The F$_{6:7}$ RIL population of 192 lines was generated by crossing K22 with DAN340, followed by subsequent self-pollination[45,46]. The K22 × DAN340 (KD) RILs and their parental lines were planted at Sanya (109.2 °N, 18.3 °E), Hainan Province, China in 2011, 2012, and 2013 and at Xinxiang (113.9 °N, 35.3 °E), Henan Province, China in 2014. Two repeats were planted at each location in 2011, 2013 and 2014, and one repeat was planted in 2012. KD RILs and their parental lines were planted in a randomized complete block design.

The association mapping panel (AMP) containing 500 diverse maize inbred lines has been reported previously[48]. The AMP lines were planted from 2011 to 2016 at three locations with a randomized complete block design. The panel lines were planted with one repeat in 2011, 2012, 2015, and 2016 and three repeats in 2013 at Sanya in Hainan Province. The lines from the same panel were planted with two repeats in two trials (i.e., Nanning (108.4 °N, 22.8 °E) in China in 2011, 2012, and Xinxiang in 2014). All lines were grown in 3.0 m row spaced 0.67 m apart with a planting density of 45,000 plants ha⁻¹.

### Disease scoring in the field
The conditions with warm temperature and high humidity in Hainan Province were favorable for SCR development. The mapping lines of RIL populations were naturally infected along with DAN340 and BY815 as the susceptible control and K22 as the resistance control. The SCR phenotype was scored at the third or fourth weeks after pollination. Using the following one-nine-point scale: "1", no disease symptom or only hypersensitivity response; "2", 5–10% leaf area infected; "3", 15–20% leaf area infected; "4", 30–40% leaf area infected; "5", 50% leaf area infected; "6", 60% leaf area infected; "7", 70% leaf area infected; "8", 80–90% leaf area infected; and "9", the whole leaf infected and dead (Supplementary Fig. 1b). The best linear unbiased predictor (BLUP) values for each RIL and AMP lines were calculated according to the methods with minor modification[46]. The BLUP values of KD RIL were used for QTL mapping by composite interval method[68].

### Inoculation in greenhouse
Maize plants were planted in the Greenhouse at Huazhong Agricultural University. The inoculation with *P. polysora* urediospores was followed by the leaf method with minor modifications[69]. Urediospores of *P. polysoda* were collected from susceptible lines in Nanbing Farm, Hainan Province. The spores were brushed down from susceptible leaves with brush pens into 20 ml distilled water with one drop of Tween-20. The spores were completely suspended and adjusted to a final concentration of $5 \times 10^5$ spores/ml. The fully extended leaves of B73, transgenic-positive and negative plants at the 6-leaf stage were painted with spore suspension. The temperature in the greenhouse was set from 25 to 30 °C. All lines were covered with non-permeable plastic films on the top to keep high humidity. The disease symptom with pinpoint spots was observable at one week after inoculation. The experiment was repeated at least three times with similar results.

### BAC library screening and sequence analysis
The BAC library of resistant line K22 was constructed at The Genome Resource Laboratory at Huazhong Agricultural University. Etiolated seedlings, grown in the dark for 2–3 weeks, were collected and frozen in liquid nitrogen. The BAC library was constructed using the standard protocols[70]. A total of 120960 clones were obtained and were saved individually in 315 384-well-plates with an average insertion size of 150 kb. Five positive BAC clones carrying *RppK* gene were identified by using PCR primers, RUST7-5 and RUST9-4 and were further confirmed with PCR amplification by primers, RRD39, RRD44, RUST8-2, RRD15, and RRD64, followed by sequencing of the PCR fragments. DNA of the five positive BAC clones were extracted using Large-Construct Kit (QIAGEN, Cat No. 12462) and was used for construction of a 20 kb insertion library for PacBio RSII. SMRT sequencing was performed in one SMRT Cell on a PacBio RSII instrument. After filtering the adaptor and low-quality reads in the raw sequencing reads, the clean data were assembled using SMRT analysis software. The contiguous sequence of about 230 kb for the genomic fragment was obtained from those BAC clones. The sequence of the target region containing three *NLR* genes, *R1*, *R2* and *R3*, were confirmed again by Sanger sequencing.

The BAC library screening and BAC clone sequencing of inbred line 1145 was performed similarly to that of K22. Two positive BAC clones covering the candidate region were identified and sequenced. The sequence results showed that the candidate region in line 1145 is 117 kb.

The BAC library of maize inbred line DAN340 was constructed from young ear tissue[70]. To optimize cost and efficiency without sacrificing time and coverage, we combined BAC clones to make superpools and after the genomic DNA fragment by partial digestion inserted them into pIndigoBAC536-S vector. Those superpools were constructed in two dimensions for each plate in multiple copied and positive clones were identified using primers: RUST7-5 and RUST9-4. The positive superpools were diluted and spread onto culture for screening of positive clones using the same primers. One positive BAC clone was identified by utilization of this strategy and was amplified by polymorphism markers to deduce the covered length. This single BAC clone from inbred line DAN340 was sequenced by Pacific Biosciences single-molecule sequencing as described above and was found to contain a 102 kb genomic fragment insertion.

The repetitive elements in the genomic sequences of candidate region were analyzed by using RepeatMasker Web Server (www.repeatmasker.org/). Gene annotation were analyzed by using the coding sequences of the B73 reference genome (AGPv2, FGS 5b[71]) with GMAP package (version 2015-09-29[72]). The gene finding programs including GENSCAN and Fgenesh were also used[73,74]. Those results were combined together manually. The sequence comparison between homology regions of *RppK* locus was done by using BLASTN suite from the National Center for Biotechnology Information. The graphical display of alignment was drawn with genoPlotR package (version 0.8.2)[75].

### Breakpoint detection in recombinant families
To confirm the breakpoints in recombinant lines between RRD103 and RRD111, they were first genotyped with primers R8.63 and R8.61 to

confirm the allele on the *R3* promoter and then tested for the presence of the *R1* and *R2* genes using primer R8.65. The results were verified by sequencing PCR products as follows: presence of both *R1* and *R2* genes, double peaks at the nucleotides 662, 676 and 694 (relative to ATG of *R2* gene) in sequencing chromatogram; presence of only *R1* gene, single peaks at those sites and "ACA" haplotype; presence of only *R2* gene, single peaks at those sites but "TGG" haplotype (Supplementary Fig. 10b, c). Based on the result of R8.63, R8.61 and R8.65, the recombination model of families was constructed. Furthermore, the R8.53 PCR products of recombination families were sequenced and were aligned with *R1*, *R2*, *R3*, and *DR3* of BAC sequences. For those families in which no breakpoint in R8.53 region was identified, they were sequenced again with R8.65F/R8.64F or R8.63R/R8.64F based on the model. Those families in which the breakpoint occurred in the last intron of *NLR* genes, they were not used for future analysis.

### Transgenic maize generation and functional validation

*R2* and *R3* genomic fragments from BAC clones were subcloned into vector pZZ01523 and the positive clones were confirmed by sequencing. The transgenic construct (pMA138) contained the genomic sequence of *R3* gene (11443 bp) including 3086 bp promoter and 2378 bp downstream regions (Fig. 1f). The transgenic construct (pMA115) contained the genomic sequence of *R2* gene (12611 bp) including 4703 bp promoter and 4464 bp downstream region (Supplementary Fig. 7a). Both pMA138 and pMA115 were transferred into KN5585, a SCR susceptible maize inbred lines, by the gene transformation platform of China National Seed Group. All the positive genomic transformation plants were self-crossed or backcrossed to KN5585. Transgenic plants were grown in Transgenic Experimental Farm of Beijing Academy of Agriculture and Forestry at Sanya, Hainan Province. Those plants including transgenic-positive plants and transgenic-negative plants were identified in each generation with *R2*- and *R3*-specific primers and were infected under natural inoculation. The procedures for SCR testing including the investigation time and standard of disease scores were the same as those used for RILs. The progeny-test assay was done in different genetic background populations. Statistically significant difference between transgenic and non-transgenic subgroups was determined by Student's *t*-test with *P*-value < 0.05.

*PPG1259* CDS region without the signal peptide (*PPG1259ΔSP*) driven by *ZmUbiquitin* promoter was cloned into pZZ01523. The positive clone was confirmed by sequencing and was transferred into KN5585 by the gene transformation platform of Weimi Biotechnology Co., Ltd.

### Cell-death assay in maize protoplasts

Maize protoplasts were isolated from 10 days old plants for transformation using PEG-calcium transfection of plasmid DNA and protoplast culture[76]. Constructs expressing candidate genes were co-transfected with the plasmid containing firefly *Luciferase* (*Luc*) gene into maize protoplasts as described before[77]. The concentrations of each plasmid and each protoplast sample were kept at the same level for all repeats and experiments. For each construct, three technical repeats were used. At 24 h after transformation, the protoplast viability was checked under fluorescence microscope (Leica). Luciferase activity was measured and quantified using a luciferase assay system (Promega). All of those experiment was repeated three times. Data were analyzed by GraphPad Prism 8 software. The experiment was repeated at least three times with similar results.

### RT-qPCR assay

Maize inbred line B73 at five-leaf stage was inoculated with *P. polysora* urediospores under high-humidity conditions. mRNA was isolated from leaf samples collected at different time points after inoculation by using Trizol (Invitrogen). Moreover, RT-qPCR was done by using

SYBR premix (TaKaRa) on real-time PCR (CFX96 Real-Time System, Bio-Rad). All of the experiments were done at least three times. Primers for detecting the expression levels of *AvrRppK* are listed in Supplementary Data 5. For each assay, the experiment was repeated at least three times with similar results.

### Single-cell DNA whole-genome sequence of *P. polysora*

For the single-cell DNA whole-genome sequencing, more than one hundred urediospores from a single uredium of *P. polysora* were picked up by needles under microscope. DNA from those spores were isolated by using the QIAGEN REPLI-g Single-Cell Kit and DNA amplification was done by MDA following the standard protocol[55]. The whole-genome amplification products were submitted for detection and sequencing.

### Identification of effector candidate genes in *P. polysora*

*P. polysora* urediospores collected from inoculated leaves in greenhouse were evenly dispersed on the surface of sterile water and kept at room temperature for 24 h. Then the mycelia were collected and washed three time by sterile water. RNA was isolated from the collected mycelia by Trizol (Invitrogen) and was submitted for Pacbio sequencing (Shanghai Personal Biotechnology Co., Ltd). The sequence results were analyzed by SMRT analysis (V2.3) and 53296 full-length sequences were identified[50]. The full-length transcripts were transcribed by Transdecoder v4.1.0 and signal peptide in ORF were analyzed by SignalP-4.0[51]. The analysis identified 965 secretory proteins and 338 of them were successfully cloned.

### Transient expression in *N. benthamiana*

For transient expression assay in *N. benthamiana*, constructs were transformed by electroporation into the *Agrobacterium* strain GV3101 and positive *Agrobacterium* transformants were identified by PCR using primers specific to each construct. *Agrobacteria* were inoculated into LB medium and grown overnight at 28 °C with shaking. *Agrobacterium* cultures were collected by centrifugation at $4602 \times g$ for 10 min and the bacterial pellet were resuspended in the infiltration buffer (10 mM MES, 10 mM MgCl$_2$ and 200 μM acetosyringone) and adjusted to OD$_{600}$ of 0.5 before infiltration into the leaves of 30-day old *N. benthamiana*. Leaves of *N. benthamiana* for transient expression. HA tagged proteins were detected by western blot using anti-HA antibody (Abcom, Cat#49969, dilution 1:1000). The experiment was repeated at least three times with similar results.

### Expression and purification of recombinant proteins in *E. coli*

For expression and purification of GST recombinant proteins in *E. coli*, the coding DNA sequences of *PPG1259* and *PPG348* without the signal peptide were cloned into the GST expression vector pGEX-4T-1. The recombinant constructs were transformed into home-made *E. coli* BL21 competent cells. The expression of recombinant proteins in *E. coli* was induced by adding isopropyl-b-D thiogalactopyranoside (IPTG) and the purification of proteins was performed according to the manufacturer's manual. The purified proteins were digested by 3C PPase to cleave the GST tag.

For the protein infiltration assay, two-week old seedlings of *RppK*-positive transgenic and negative transgenic plants from the same transgenic line were used. Purified PPG1259ΔSP and PPG348ΔSP proteins were diluted to 0.2 mg/ml in 1× PBS buffer and infiltrated into the first leaf of *RppK*-positive and negative transgenic plants using 1 ml syringe. The hypersensitive response (HR) was evaluated at 24 h after infiltration. The experiment was repeated at least three times with similar results.

### ROS detection assay and MAPK kinase activity assay

For ROS detection assay, two independent *PPG1259ΔSP* transgenic lines at three-leaf stage were tested in this assay. Leaf discs were taken

by using a 4 mm diameter punch. All discs were put in 1% DMSO for overnight, then 1% DMSO was replaced by 2× L-012 in 0.005% Silwet L-77 and disc samples were treated for 1 h. Two kinds of 2× HRP buffer were prepared; one was for the treatment buffer containing 50 mg/ml chitin and the other was for the mock control without chitin. Next, eight cells were treated with treatment buffer and the others were treated with the mock buffer. A microplate detector was used to collect light signals, which reflected ROS production. The signal was read every one minute for one hour and the data were analyzed to display the changes of ROS production over time. The experiment was repeated at least three times with similar results. Moreover, statistical tests were done by two-tailed Student's test with $P < 0.05$.

For chitin-induced MAPK kinase activity assay, two independent *PPG1259ΔSP* transgenic lines at six-leaf stage were tested in this assay. The fully extended leaves were treated with 50 mg/ml chitin for 0, 5, 15, or 30 min. Then leaf samples were grounded for protein extraction. Equal amount of proteins was loaded onto a 12% SDS-PAGE gel for western blot. MAPK kinase activity was examined by using anti-phospho-p44/42 antibody (Cell Signaling Technology, 9101S, dilution 1: 1000) and the loading actin control was examined by using anti-Actin antibody (ABclonal, AC009, dilution 1: 5,000). Relative MAP kinase activity was normalized to the level of Actin protein. The experiment was repeated at least three times with similar results. Statistical tests were done by two-tailed Student's *t*-test with *P* value < 0.05.

### Molecular breeding of *RppK*

The cultivar K22 carrying *RppK* was crossed with two parental inbred lines of JK968 (Jing724 × Jing92), an elite hybrid widely planted in China, and backcrossed four time with recurrent parents to generate the BC$_4$F$_1$ plants, which self-crossed two time to generate the BC$_4$F$_3$ lines. Molecular markers were used to maintain the K22-derived *RppK* locus. Two pairs of near isogenic lines (NILs): Jing724$^{RppK}$ and Jing724, Jing92$^{RppK}$ and Jing92 were selected from the BC$_4$F$_3$ progenies. Selected Jing724$^{RppK}$ and Jing92$^{RppK}$ carrying the *RppK* locus were genotyped with Genotyping-by-Sequencing and found to contain 97.4% and 96.9% of their respective recurrent parent genome. The Jing724$^{RppK}$ and Jing92$^{RppK}$ lines were crossed to generate the improved hybrid JK968a (Jing724 × Jing92$^{RppK}$) and JK968b (Jing724$^{RppK}$ × Jing92$^{RppK}$). The improved IL4$^{RppK}$ line was selected with similar molecular assisted selection. The inbred line IL3 was crossed with IL4 or improved IL4$^{RppK}$ to generate the original Hybrid2$^{wt}$ or the improved Hybrid2$^{RppK}$ (IL3 × IL4$^{RppK}$). The inbred lines: Jing72464, JingMC01 and Jing88 were respectively crossed with inbred line Jing2416 and Jing2416K (with the same *RppK* locus as K22), respectively, to generate paired hybrids: MC121$^{wt}$ and MC121$^{RppK}$, JNK728$^{wt}$ and JNK728$^{RppK}$, JNK828$^{wt}$ and JNK828$^{RppK}$.

To test the performance of hybrids in the field, we planted those hybrids in fields for yield tests under two conditions: a) none-SCR condition: fields in Tongzhou (116.6 °N, 39.9 °E), Changping (116.2 °N, 40.2 °E) and Huanggang (114.9 °N, 30.5 °E), China, in 2020; b) SCR condition: natural SCR nursery fields in Bozhou (115.8 °N, 33.9 °E) and Yongcheng (116.4 °N, 33.9 °E), China in 2020. Field trials of hybrids JK968 and JK968a were conducted in three locations: Bozhou, Tongzhou and Changping. Field trials for hybrids MC121$^{wt}$ and MC121$^{RppK}$, JNK728$^{wt}$ and JNK$^{RppK}$, JNK828$^{wt}$ and JNK828$^{RppK}$ were done in two locations: Bozhou and Tongzhou. Each line was planted in nine subplots and each subplot contains two rows. Each row was 5 m long and spaced 60 cm with 19 plants. Field trials of hybrids JK968 and JK968b, Hybrid2$^{wt}$ and Hybrid2$^{RppK}$ were conducted in two locations: Yongcheng and Huanggang. Each line was planted in sixteen subplots and each subplot contains one row. Each row was 3 m long and spaced 50 cm with 11 plants. The agronomic traits (flowering time, plant height and ear height) of each line were investigated. All ears were harvested and dried to uniform moisture for scoring ear and grain traits. Yield-related traits (ear length, ear diameter, ear weight and kernel weight per plant) in a subplot were calculated. Statistically significant

difference between wild-type lines and improved lines was evaluated by two-tailed Student's *t*-test with *P*-value < 0.05.

The transgenic-positive line KN5585$^{RppK}$ and negative line KN5585$^{Wt}$ were crossed with four inbred lines (B73, Mo17, IL1, and IL2) to generate F$_1$ hybrids. Those hybrids were planted in Sanya, China in 2019 to test the yield performance. Each hybrid line was planted eight rows and each row was set as one subplot. The SCR score and yield-relative traits were investigated. Statistically significant difference between hybrid line without *RppK* gene and improved hybrid lines with *RppK* gene was evaluated by two-tailed Student's *t*-test with *P* value < 0.05.

### Reporting summary

Further information on research design is available in the Nature Research Reporting Summary linked to this article.

## Data availability

The *RppK* locus genomic DNA sequences from K22, DAN340, and 1145 are available at NCBI under accessions MZ322317, MZ322318, MZ312612, respectively. The RNA-seq data generated by PacBio sequencing is available at NCBI under the BioProject ID PRJNA732947; the *P. polysora* genomic DNA sequence data generated by single cell is available under the BioProject ID PRJNA732557. Source data are provided with this paper.

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

## Acknowledgements

We thank Dr. Mingling Xu at China Agricultural University, Dr. Zhixiang Chen at Purdue University, Dr. Jeffery L. Dangl at University of North Carolina, Dr. Hanhui Kuang at Huazhong Agricultural University, and Dr. Xuewei Chen at Sichuan Agricultural University for critical reading and comments. This work was supported by The National Key Research and Development Program of China (2016YFD0101002, Z.L.), the National Natural Science Foundation of China (31571676, Z.L.; 31761143008, Z.L.; 32072007, Z.L; and 31161140347, J.Y.).

## Author contributions

G.C., H.W., and J.D. cloned *RppK* gene by map-based strategy and did disease evaluation. B.Z. did *P. polysora avr* gene screening, identified *AvrRppK* gene and examined the function of *AvrRppK* in suppressing PTI. B.Z., J.W., and H.Z. did protoplast assays. Q.P. sequenced *AvrRppC* gene in different *P. polysora* isolates and checked *RppK* genotypes in different hybrid lines. Junshi Huang, Jiabao Hou, Junfei Dong, J.Q., and F.Z. cloned genes encoding *P. polysora* secretory proteins. Q.Y. identified and cloned *RppK* gene transcripts and cloned *RppS* gene. G.C., H.W., and H.M. did bioinformatic analysis. W.L. did maize breeding and field management. C.D. did genotype analysis. G.T. and B.L. repeated the function of *AvrRppK* gene. X.L. generated maize transgenic plants. G.C., R.Z., and J.Z. evaluated the agricultural traits of hybrids. G.C. J.Y., and Z.L. wrote the manuscript. J.Y. and Z.L. directed the project.

## Competing interests

The authors declare the following competing interests:
Patents with application No. PCT/US20 19/032497 (Methods of identifying, selecting, and producing southern corn rust resistance crops; Inventors: J.Y.; B.L.; G.T.; G.C.; J.D.; Z.L.; H.W.; Q.Y.) and WO 2019/ 236257 (Plant pathogen effector and disease resistance gene identification; Inventors: Z.L.; B.Z.; J.W.; Junshi Huang; H.W.; H.M.) are pending. Other authors claim no competing interests.
