## [Peer Review File · Nature Communications]

RppK Mediates Maize Broad Resistance against Southern Corn Rust through Its Cognate Gene AvrRppKReviewers' Comments:

Reviewer #1:

Dear editor,

This manuscript provides significant amount of genetic evidences to support the first identification of a pair of a core effector and its cognate NLR that defends maize against southern corn rust. Chen and coauthors identified the NLR RppK based on genetic mapping, identified AvrRppK based on sequencing and bioinformatic analysis, and further showed that recognition of AvrRppK by RppK provides value agronomic use in maize breeding. In general, this manuscript provides clear and solid evidences in showing that RppK recognizes the core effector AvrRppk to confer resistance. However, efforts need to be made to deal with the following minor revisions.

1. Line 49 writes "ETI activates more rapid and stronger immunity than PTI". Please cite references claiming ETI is more rapid than PTI.
2. Core effectors are required for virulence in crops (line 67). So, is AvrRppK required for pathogenesis?
3. Line 85, it would be nice to introduce CC-type NLR at somewhere, such as line 49.
4. R2 fragment for vector construction is 11.6 kb (line 131) or 12.6 kb (fig. S7)?
5. Please provide evidences supporting positive and negative lines in Fig 1.
6. Fig. S1, is BY815 necessary or introduced?
7. Fig. S2, why S and R is used only for RppK?
8. In Fig. 2, please provide field performance for tested lines in a format similar to Fig. 2B. In addition, please keep similar data amount for JK968a and JK968b by re-arranged Fig. 2C and D together with Fig. S10.
9. Fig. 2E legend, add JK968b. In addition, how about the grain yield of JK968wt, JK968a and JK968b without infection? Vice versa for TZ, CP and HG.
10. 724RppK instead of 724 was chosen to make JK968b. So, does RppK in 724 background contribute to resistance?
11. Does recognition of AvrRppK by RppK confers resistance against all *P. polysora* strains?
12. Line 163, is "These SCR-resistant JK968 lines are currently commercially cultivated in China" informatic? If not, please remove it.
13. Line 173-184, so RppS is RppK? Does RppS recognize AvrRppK and trigger HR in *N. benthamiana*?
14. Fig. 3B, please show accumulation of expressed proteins, and show representative LUC signal for each group. In addition, please explain why Rp1-D21 makes a difference in LUC level between RppK transgenic and non-transgenic plants? What is more, the error bar referring to EV in RppK non-transgenic plants appears to be wider than others.
15. Fig. 3C, please align infiltration sites and dot circles, and show protein accumulation. Fig. 3D is over-tagged, please either remove RppK transgenic plants or RppK+/-, and show accumulation of purified proteins.
16. AvrRppK is conserved among strains collected from different part of China. Since AvrRppK is claimed as a core effector, it would be nice to check the similarity of

AvrRppK among *P. polysora* strains from other countries, and to check the conservation level among fungi as well as other pathogens.

17. Fig. 4C, is INF1 also conserved in *P. polysora*? Otherwise it would be better to use *P. polysora*-derived PAMPs.
18. Line248-255, the observation that AvrRppK is able to suppress INF1-triggered HR is not robust enough to support the conclusion that AvrRppK is a virulence effector. Please provide genetic evidences, such knocking down/out of AvrRppK reduces pathogenesis or overexpression of AvrRppK promotes infection, to support this conclusion.
19. Since the core effector AvrRppK is recognized by RppK that is not prevalently distributed in maize, a comprehensive discussion on the arm-race aspect between core effectors and cognate NLRs would be appreciated.
20. To strengthen the practicable value of RppK, it is appreciated to test whether RppK/AvrRppK-mediated immune responses could be suppressed by known intracellular effectors or not, and to test whether RppK-mediated resistance could be bypassed by strains from other regions.
21. Format of references should be intensively checked for capitalization, italicization and abbreviation.
22. Scientific writing should be comprehensive improved in general.

Reviewer #2:

Remarks to the Author:

The manuscript by Chen et al., reports the identification of a novel RppK of CNL resistance protein from the maize inbred line K22 and its cognate avirulent effector AvrRppK from *Puccinia polysora*. They cloned RppK gene through map-based cloning and showed the introgression of the RppK gene into different maize inbred lines and hybrid lines enhanced resistance against *P. polysora*. Furthermore, authors identified AvrRppK protein recognized by RppK through protoplast transient assay and showed that co-expression of RppK and AvrRppk in *Nicotiana benthamiana* induced hypersensitive response. Interestingly, they found that AvrRppK gene was highly conserved in 38 isolates of *P. polysora*, while the RppK gene was relatively rare in maize germplasm. This makes the RppK gene potentially valuable as it will likely confer a novel, broad resistance against southern rust. Overall this is an impressive piece of work and will be of significant value to the field.

We do have a few questions/issues though, which focus around the claim that AvrRppK is a "core effector".

All analysed isolates of *P. polysora* carried the identical AvrRppK sequence. Did the authors inoculate any of these isolates on maize lines carrying RppK? Did they confer resistance against all isolates of *P. polysora*?

Variation among pathogen isolates is crucially dependent on the host plants from which they were collected. We have to assume that none of the host plants from which the ~100 *P. polysora* isolates were collected carried RppK, is that right? Did they have other SCR resistance genes? Ideally, in order not to bias the results, the host plants from which the isolates were collected should have no known resistance alleles.

Do you ever see any SCR on maize line K22? If so do those isolates carry AvrRppK?

We think that it is premature to conclude that AvrRppk is a virulence factor just based on the result that AvrRppK suppresses INF1-induced cell death in *N. benthamiana*. This is certainly an interesting piece of evidence, but it is, in the end, evidence that it suppresses the effects of a response in a foreign host to a protein that does not occur in the maize/*P. polysora* system- so it is interesting but not directly relevant.

The authors should test a *P. polysora* knock out mutant which does not express AvrRppK in maize lines if they want to verify that AvrRppK is a virulence factor. In the absence of this evidence, they should be a lot more circumspect in their interpretation of their results

The definition of a "core effector" that the authors use is vague but it comes down to 2 things:

1. The effector is widespread in pathogen isolates
 2. The effector is a virulence factor (a criterion which we think would apply to most effectors)
- Not having any evidence to the contrary, we have to assume that AvrRppk is not under the selective pressure in the absence of RppK being deployed in the field. The fact that AvrRppK is widely found is very likely caused by a combination of the rarity of RppK among maize germplasm and possibly the way the authors conducted their sampling.

If RppK were deployed in a widespread manner, do the authors think that the frequency of AvrRppK would decrease? If so, then we would argue it is just a regular effector, not a "core effector".

So we believe that the authors should be more careful in their claims about RppK being a core effector sensor and its ability to confer broad spectrum resistance. The fact is that, as with any R-gene, until it is deployed on a wide scale we don't really know how broad spectrum or durable it will be.

We don't think that the title. "Deployment of a core effector sensor, RppK, in maize confers broad spectrum resistance against southern corn rust", gives the reader a good idea of what was actually done- The paper describes identification of both a resistance gene (not a "core effector sensor") and the corresponding Avr gene. We think the title should say this!

None of this detracts from the really impressive amount of work reported or the interesting results.

Minor comments:

In line 125, you have to add 4H1028 in brackets.

In line 131, R2 is 12.6 kb, not 11.6.

In line 134, KN5585 is a susceptible maize line? If so, you need to give the simple information about it.

In line 163, 'These SCR-resistant JK968 lines are currently commercially cultivated in China.' should be deleted. You mentioned it in line 157 already.

In line 799, JK968b should be added next JK968a. Fig2 (E) has the data with JK968b.

'JK968b was derived from a cross between Jing724RppK and Jing92RppK (Fig. S10)' should move to (E) legend.

In figure 3C, there is no positive control in *N benthamiana*. You can infiltrate Rp1-D21 which you used in luciferase assay.

In supplementary data, what is 1145 in fig S3? Why did you use this line?

In Fig S8, (B) has four gel pictures. In Del13K marker, is it correct to write 13K, not DR3? Please confirm this.

Reviewer #3:

Remarks to the Author:

This study describes the cloning of a disease resistance gene (designated RppK) and that of its corresponding avirulence gene (AvrRppK) in the maize-Southern corn rust (SCR) pathosystem. SCR is a serious disease of maize that seems to be getting worse and worse perhaps because of climate change or agronomic practices. Many maize genes and QTL conferring resistance to SCR have been defined by genetic studies but none has been cloned thus far. So, this represents the first report of the successful cloning of an R gene that confers race-specific resistance to SCR.

RppK was found in a Chinese inbred K22, in which the authors claim the gene has been successful in conferring resistance over the past 30 years. On the basis of this observation they claim it to be a durable gene. A map based cloning approach was used to clone a candidate gene for RppK, which was subsequently validated by transgenesis to be the correct one. In addition, they showed that another inbred line SCML205 that was previously characterized to contain the RppS resistant locus actually contains an allele of RppK in that their genic sequence is identical except for an intronic 2 bp indel in RppS. No surprise that the gene encodes a typical NLR, a CNL.

RppK was transferred to a number of susceptible elite inbreds, and the hybrids generated from them were shown to have superior yields in the presence of the disease and no yield penalty in its absence. After having cloned RppK, the authors sought to clone the SCR effector gene that RppK intercepts to confer HR. The authors used an elegant approach to accomplish it. In brief, they first sequenced the genome of the rust pathogen (*Puccinia polysora*; PP, an isolate from Wuhan) and bioinformatically identified 965 genes predicted to encode host secreted proteins. Three hundred and thirty-eight of these genes were cloned and tested in a transient protoplast assay system to detect cell death (HR). Only one gene – designated PPG1259 – triggered a robust HR and was considered a candidate gene for AvrRppK. They next used the *Nicotiana benthamiana* heterologous system to show that co-infiltration of RppK and PPG1259 does indeed trigger an HR. Another assay that they used to provide further support to the correct cloning of the Avr gene, was based on the injection of purified proteins of PPG1259 and that of a putative effector gene PPG348 as a control in plants containing and lacking transgenic RppK. The observation that only CML1259 caused cell death and that too in the plant containing the RppK transgene was deemed as confirmation that PPG1259 is AvrRppK. It encodes a 96-aa protein that exhibits no sequence identity with any known proteins or domains.

Next, they showed that the genotype of the Avr1259 gene in more than 100 isolates of SCR that they collected from three different provinces (Hainan, Guaxi and Hubei) was the same. The gene was then amplified and sequenced from 37 isolates (from all three provinces) and found to have the identical sequence. This high conservation of sequence was interpreted to mean that this gene represents a core effector and thus resistance against it is expected to be durable.

Overall it is a decent manuscript, and I am convinced that the authors have definitely cloned the gene underlying RppK and perhaps also its corresponding avirulence gene, AvrRppk. However, there are a few concerns that I would like to bring up here.

First, I am not sure if the evidence is there yet to suggest that AvrRppk is a core effector. Sure, they did show that there is a great deal of sequence conservation, but they provided no evidence of the race structure of these isolates. One possibility is that there is very little diversity in SCR in China. It would have been better if they also sequenced a couple of other effector genes in addition to PPG1259 to address their conservation.

The experiment they did to show that AvrRppK has virulence activity is also concerning. They used suppression of cell death mediated by INF1, which they say is a typical PAMP, as an assay for the virulence activity of AvrRppK. I don't think induction of cell death is a typical PTI response.

Regardless, it would have been more convincing if they also looked at the effect of AvrRppK on Rp1-D21-mediated cell death.

I could not understand the rationale for doing the single-cell sequencing technology experiment, what they got out of it, and why it matters for this work.

I would have appreciated if the authors discussed the similarities, if any, in the ancestry of the 17 lines of the diversity panel. Are any of these lines from places other than China? How prevalent was the RppK gene in the commercial germplasm? I think these questions are relevant given the claim that this R gene has been durable for more than 30 years.

Reviewer#1

This manuscript provides significant amount of genetic evidences to support the first identification of a pair of a core effector and its cognate NLR that defenses maize against southern corn rust. Chen and coauthors identified the NLR RppK based on genetic mapping, identified AvrRppK based on sequencing and bioinformatic analysis, and further showed that recognition of AvrRppK by RppK provides value agronomic use in maize breeding. In general, this manuscript provides clear and solid evidences in showing that RppK recognizes the core effector AvrRppk to confer resistance. However, efforts need to be made to deal with the following minor revisions.

- 1. Line 49 writes “ETI activates more rapid and stronger immunity than PTI”. Please cite references claiming ETI is more rapid than PTI.**

Answer: Thank you for your comments. We added two references to claim ETI is more rapid than PTI.

References:

- (1) Mine et al. 2018. The Defense phytohormone signaling network enables rapid, high-amplitude transcriptional reprogramming during effector-triggered immunity. *The Plant Cell*. 30: 1199-1219.
- (2) Yuan et al. 2021. PTI-ETI crosstalk: an integrative view of plant immunity. *Current Opinion in Plant Biology*. 62: 102030.

- 2. Core effectors are required for virulence in crops (line 67). So, is AvrRppK required for pathogenesis?**

Answer: Thank you for your comments.

We generated transgenic maize plant overexpressing *AvrRppK ΔSP* driven by the *ZmUbi* promoter. After inoculation with *P. polysora*, transgenic positive plants showed more susceptible phenotype to *P. polysora* than transgenic negative plants in two independent lines. It indicates overexpression of *AvrRppK ΔSP* in maize can enhance plant susceptibility to SCR.

By using transgenic plants overexpression *AvrRppK ΔSP*, we checked the chitin-triggered immunity (MAP kinase activity and ROS accumulation). The results showed that transgenic positive plants showed weaker chitin-triggered MAP kinase activity and ROS accumulation than transgenic negative plants. It indicates that overexpression of *AvrRppK ΔSP* in maize can suppress chitin-triggered immunity. All of these results indicate that AvrRppK is a virulence factor.

Although we confirmed the virulence function of AvrRppK, we still do not know whether it is required for the pathogenicity. To examine its pathogenicity, *avrppk* mutants of *P. polysora* are required. *P. polysora* is a biotrophic pathogen. Up to now, there is no way to do genetic modification in *P. polysora*. Another way is to overexpressing *AvrRppK*-RNAi construct in maize. We made *AvrRppK*-RNAi construct and generated *AvrRppK*-RNAi transgenic maize lines. However, many transgenic plants showed bleached leaves in *AvrRppK*-RNAi transgenic maize lines and they died before four-leaf stage. And, we could not test their disease phenotype to SCR. So, we do not know whether AvrRppK is required for pathogenicity, although it is a virulence factor.

- 3. Line 85, it would be nice to introduce CC-type NLR at somewhere, such as line 49.**

Answer: Thank you for your comment. One sentence was added on line 51-53: Based on their N-terminal domains, NLRs have been classified into two major groups: the coiled-coil-NLRs (CC-NLRs) and Toll/interleukin-1 receptor/Resistance-NLRs (TIR-NLRs).

4. R2 fragment for vector construction is 11.6 kb (line 131) or 12.6 kb (fig. S7)?

Answer: Thank you for your comment. It is my fault.

R2 fragment cloned into vector is 12.6 kb. It has been corrected in line 135.

5. Please provide evidences supporting positive and negative lines in Fig 1.

Answer: Thank you for your comments. In Fig S8A, we genotyped the transgenic plants with molecular markers R8.63 and R8.61 to support positive and negative lines in Fig. 1.

R8.83 is specific to *R3* gene. A specific PCR fragment can be amplified from *RppK* transgenic positive plants and no PCR fragment can be amplified in *RppK* transgenic negative plants. Molecular marker R8.61 was used to check DNA quality. Because R8.61 is specific to *DR3* gene which is broadly attributed in maize inbred lines except K22. A PCR fragment can be amplified from all *RppK* transgenic positive and negative plants, but no PCR fragment can be amplified from K22.

In Fig S8B, we did qRT-PCR to check *R3* gene expression in *RppK* transgenic positive plants and transgenic negative plants of two independent families.

6. Fig. S1, is BY815 necessary or introduced?

Answer: Thank you for your comments.

BY815 is a maize inbred line very susceptible to southern corn rust and it was used as a control. The information about BY815 was added into the figure legend for Fig. S1.

7. Fig. S2, why S and R is used only for *RppK*?

Answer: Thank you for your comments. I am sorry that we did not label it clearly.

Here, we tried to label the SCR disease phenotype of those key recombinants in Fig. S2 and “*RppK*” was a wrong label. We changed “*RppK*” into “Phenotype”, and put it on the bottom of the table. In this table, “S” means susceptible to SCR and “R” means resistant to SCR.

8. In Fig. 2, please provide field performance for tested lines in a format similar to Fig. 2B. In addition, please keep similar data amount for JK968a and JK968b by re-arranged Fig. 2C and D together with Fig. S10.

Answer: Thank you very much for your comments.

We added the field performance for JK968^{wt} and JK968a and that for JK968^{wt} and JK968b in Fig. 2; and we moved the figures of Fig.S10 into Fig. 2.

9. Fig. 2E legend, add JK968b. In addition, how about the gain yield of JK968wt, JK968a and JK968b without infection? Vice versa for TZ, CP and HG.

Answer: Thank you very much for your comments. I am sorry that we did not describe it clearly.

The SCR disease phenotypes of JK968^{wt}, JK968a and JK968b are exhibited in Fig. 2E. 20BZ means the field in Bozhou in 2020, 20YC means the field in Yongcheng in 2020, 20TC means the field in Tongzhou in 2020, 20CP means the field in Changping in 2020, and 20HG means the field

in Huanggang in 2020. SCR occurred in the fields of 20BZ and 20YC, but did not occur in the fields of 20TC, 20CP and 20HG.

In Fig. 2F, the yield data of JK968a, JK968b and JK968^{wt} under SCR disease conditions (20BZ and 20YC) were exhibited on the left side of the dash line; and the yield data of JK968a, JK968b and JK968^{wt} without SCR disease (20TC, 20CP and 20HG) were exhibited on the right side of the dashed line.

In Fig. 2F, there was no SCR disease in 20TZ, 20CP and 20HG. Based on the results in Fig. 2F, there was no difference on the grain yield between JK968^{wt} and JK968a in the field of 20TZ; and there was no difference on grain yield between JK968^{wt} and JK968b in the fields of 20CP and 20HG.

10. 724Rppk instead of 724 was chosen to make JK968b. So, does *RppK* in 724 background contribute to resistance?

Answer: Thank you for your comment.

Yes, *RppK* in Jing724 background contributed resistance to SCR. Jing724 is susceptible to SCR (Fig2A). Jing724^{RppK} line contains *RppK* gene in Jing724 background (Fig2A).

11. Does recognition of AvrRppK by *RppK* confers resistance against all *P. polysora* strains?

Answer: Thank you for your comments. We do not think *RppK* confers resistance against all *P. polysora* strains.

We planted K22 in different areas in China and K22 showed highly resistant phenotype to SCR in all tested areas. Also, we inoculated the transgenic plants containing the *RppK* genomic DNA fragment with five different isolates from different areas and observed that transgenic positive plants showed more resistant phenotype to SCR than negative transgenic plants (Fig. S9). So, our conclusion is that recognition of AvrRppK by *RppK* confers broad resistance against *P. polysora*.

Since we only did inoculation on transgenic plants with five isolates, we cannot conclude that *RppK* contributes resistance against all *P. polysora* isolates.

12. Line 163, is “These SCR-resistant JK968 lines are currently commercially cultivated in China” informatic? If not, please remove it.

Answer: Thank you for your comment. We deleted this sentence.

We introgressed *RppK* gene from K22 into multiple hybrid lines and some of these SCR-resistant lines are currently commercially cultivated in China. But those companies refused to give us any official verification, because it is a trade secret for any company who used *RppK* gene for breeding.

13. Line 173-184, so *RppS* is *RppK*? Does *RppS* recognize AvrRppK and trigger HR in *N. benthamiana*?

Answer: Thank you for your comments.

Yes, after infiltrating *RppS* with AvrRppK in *N. benthamiana*, we did observe clear HR (Fig. S20). So, *RppS* can recognize AvrRppK and trigger HR in *N. benthamiana* (Line 240-line 246).

14. Fig. 3B, please show accumulation of expressed proteins, and show representative LUC signal for each group. In addition, please explain why *Rp1-D21* makes a difference in LUC

level between *RppK* transgenic and non-transgenic plants? What is more, the error bar referring to EV in *RppK* non-transgenic plants appears to be wider than others.

Answer: Thank you very much for your comments.

We repeated this experiment again and checked the protein levels of Rp1-D21-3 × HA, PPG1259 Δ SP-3 × HA (Fig.S19). In Fig. S19, “*” means Rp1-D21-3 × HA protein, “***” means PPG1259 Δ SP-3 × HA.

In the last version of the manuscript, *Rp1-D21* makes a difference in LUC level between *RppK* transgenic and non-transgenic plants. The problem was caused by the transformation efficiency. *RppK* transgenic and non-transgenic plants are two different materials. Transient expression of *Rp1-D21* in protoplasts can cause strong cell death, and LUC was used to indicate the survival cells. So, a small difference on the transformation efficiency could cause big difference on the number of survived cells. And the error bar referring to EV appears to be wider than others. This problem might be caused by the quality of EV plasmid.

In order to solve the two problems, we isolated new plasmids and repeated this experiment. Please check Fig. 3B. As shown in Fig 3B, *Rp1-D21* and *EV* did not make a difference in LUC level between *RppK* transgenic and non-transgenic plants.

15. Fig. 3C. please align infiltration sites and dot circles, show protein accumulation. Fig. 3D is over-tagged, please either remove *RppK* transgenic plants or *RppK*+/-, and show accumulation of purified proteins.

Answer: Thank you for your comments.

For Fig. 3C, we repeated this experiment, *Rp1-D21* construct was infiltrated as the positive control, and the infiltration sites were labelled by “*”. The protein levels of PPG1259 Δ SP-3 × HA and Rp1-D21-3 × HA were shown in Fig. S20B. Since we infiltrated *R3* genomic DNA in *N. benthamiana* with *PPG1259 Δ SP-3 × HA*, we did qRT-PCR to check the expression levels of *RppK* gene in different infiltrations (Fig. S20A).

For Fig. 3D, we removed *RppK*+/- . And the purified protein levels of PPG1259 Δ SP and PPG348 Δ SP were presented in Fig. S21C.

16. *AvrRppK* is conserved among strains collected from different part of China. Since *AvrRppK* is claimed as a core effector, it would be nice to check the similarity of *AvrRppK* among *P. polysora* strains from other countries, and to check the conservation level among fungi as well as other pathogens.

Answer: Thank you for your comments.

We tried to contact with researchers in American, but we did not get response from them. I also checked the SCR information in American. Normally, SCR happens from June to December in American. So, there was no *P. polysora* spores in the field during these days (From this January to this April). Maybe, that was the reason why those researchers did not give us responses. So, we did not check the similarity of *AvrRppK* in *P. polysora* strains from other countries.

We also did BLAST analysis to check *AvrRppK* homolog genes from other pathogens in NCBI in the DNA level and the protein level. But we did not get any hit. It indicated that *AvrRppK* might only exist in *P. polysora*.

17. Fig. 4C, is INF1 also conserved in *P. polysora*? Otherwise it would be better to use *P.*

***polysora*-derived PAMPs.**

Answer: Thank you for your comments.

INF1 is not from *P. polysora*. And we deleted this data from Fig. 4. Up to known, there is no *P. polysora*-derived PAMP was reported. So, we use fungus-derived PAMP, chitin, to check PTI on maize.

In order to check whether AvrRppK suppresses PTI in maize, we generated transgenic maize plants overexpression *AvrRppK* Δ SP. Then those transgenic plants were treated with chitin, which is a typical fungal PAMP to induce PTI.

After treatment with chitin, strong PTI responses (MAP kinase activity and ROS accumulation) were activated in transgenic negative plants. While, PTI responses (MAP kinase activity and ROS accumulation) in transgenic positive plants overexpression *AvrRppK* Δ SP were weaker than these in transgenic negative plants (Fig.4G-4H and Fig. S25).

These results indicates that expression of *AvrRppK* can suppress PTI in maize.

18. Line 248-255, the observation that *AvrRppK* is able to suppress INF1-triggered HR is not robust enough to support the conclusion that *AvrRppK* is a virulence effector. Please provide genetic evidences, such knocking down/out of *AvrRppK* reduces pathogenesis or overexpression of *AvrRppK* promotes infection, to support this conclusion.

Answer: Thank you for your comments. I agree with you that the observation that AvrRppK suppressed INF1-mediated cell death is not enough to support that AvrRppK is a virulence effector.

In order check whether AvrRppK is a virulent effector or not, we generated transgenic maize plants overexpressing *AvrRppK* Δ SP.

After inoculation with *P. polysora*, transgenic positive plants in two independent lines showed more susceptible phenotype to SCR than transgenic negative plants. It indicates that AvrRppK can enhance plant susceptibility to SCR (Fig. 4D-4F).

Also, we checked whether AvrRppK can suppress PTI by using these transgenic plants. After chitin treatment, strong PTI responses (MAP kinase activity and ROS accumulation) were activated in transgenic negative plants. While MAP kinase activity and ROS accumulation triggered in transgenic plants overexpressing *AvrRppK* Δ SP were much weaker than these in transgenic negative plants (Fig.4G-4H and Fig. S25).

So, we concluded that AvrRppK is a virulent factor and its expression in maize can suppress chitin-triggered PTI responses.

19. Since the core effector *AvrRppK* is recognized by *RppK* that is not prevalently distributed in maize, a comprehensive discussion on the arm-race aspect between core effectors and cognate *NLRs* would be appreciated.

Answer: Thank you for your comments.

Yes, *RppK* is not prevalently distributed in maize might explain why no polymorphism was detected in *AvrRppK*. To figure out this puzzle, more work on core effectors and their corresponding *NLRs* should be done. And we added a short discussion in line 375-line 391.

20. To strengthen the practicable value of *RppK*, it is appreciated to test whether *Rppk/AvrRppK*-mediated immune responses could be suppressed by known intracellular effectors or not, and to test whether *RppK*-mediated resistance could be bypassed by

strains from other regions.

Answer: Thank you for your comments.

Up to now, only one effector (*AvrRppC*) was reported in *P. polysora* (Deng et al, 2022, Molecular Plant). And K22, the donor of *RppK* gene, showed highly resistant phenotype to isolate *PP.Hainan* and isolate *PP.CN1.0* which contain *AvrRppC* (Fig. S9), although K22 does not contain *RppC* gene (Deng et al. 2022. Molecular Plant). It indicates that *RppK*-mediated resistance cannot be suppressed by *AvrRppC*.

We inoculated transgenic plants containing *RppK* genomic DNA sequence with five different *P. polysora* isolates from Henan, Guangdong and Hainan. Two of them (*PP.CN1.0* and *PP.Hainan*) contains *AvrRppC^{ref}*. And we observed that transgenic positive plants containing *RppK* showed highly resistant phenotype to all five *P. polysora* isolates; while, transgenic negative plants showed highly susceptible phenotype to all five *P. polysora* isolates. The results confirmed that *AvrRppC* cannot suppress *RppK*-mediated resistance.

Reference: Deng et al., The *RppC*-*AvrRppC* NLR-effector interaction mediates the resistance to southern corn rust in maize, *Molecular Plant* (2022), <https://doi.org/10.1016/j.molp.2022.01.007>

21. Format of references should be intensively checked for capitalization, italicization and abbreviation.

Answer: Thank you very much for your comments.

We checked all references and corrected their format.

22. Scientific writing should be comprehensive improved in general.

Answer: Thank you for your comments.

We sent the manuscript to “Springer Nature Author services (SNAS)” and the manuscript was edited for proper English language, grammar, punctuation, spelling, and overall style by one or more of the highly qualified native English speaker editors at SNAS, which could be verified on the SNAS website using the verification code 8E89-C6B9-CB33-A29C-492P.

Reviewer#2

The manuscript by Chen et al., reports the identification of a novel **RppK** of CNL resistance protein from the maize inbred line K22 and its cognate avirulent effector **AvrRppK** from **Puccinia polysora**. They cloned **RppK** gene through map-based cloning and showed the introgression of the **RppK** gene into different maize inbred lines and hybrid lines enhanced resistance against *P. polysora*. Furthermore, authors identified **AvrRppK** protein recognized by **RppK** through protoplast transient assay and showed that co-expression of **RppK** and **AvrRppk** in *Nicotiana benthamiana* induced hypersensitive response. Interestingly, they found that **AvrRppK** gene was highly conserved in 38 isolates of *P. polysora*, while the **RppK** gene was relatively rare in maize germplasm. This makes the **RppK** gene potentially valuable as it will likely confer a novel, broad resistance against southern rust.

Overall this is an impressive piece of work and will be of significant value to the field.

We do have a few questions/issues though, which focus around the claim that **AvrRppK** is a “core effector”.

- 1. All analysed isolates of *P. polysora* carried the identical *AvrRppK* sequence. Did the authors inoculate any of these isolates on maize lines carrying *RppK*? Did they confer resistance against all isolates of *P. polysora*?**

Answer: Thank you for your comments.

Yes, we did inoculate five isolates on maize transgenic lines carrying *RppK*. And transgenic positive plants in two independent lines showed highly resistant phenotype to the five tested isolates; while, transgenic negative plants showed highly susceptible phenotype to the five tested isolates (Fig. S9).

- 2. Variation among pathogen isolates is crucially dependent on the host plants from which they were collected. We have to assume that none of the host plants from which the ~100 *P. polysora* isolates were collected carried *RppK*, is that right? Did they have other SCR resistance genes? Ideally, in order not to bias the results, the host plants from which the isolates were collected should have no known resistance alleles.**

Answer: Thank you very much for your comments. I agree with you that the isolates ideally should be collected from plants which have no known resistance alleles.

Up to now, only two resistance genes (*RppC* and *RppK*) have been cloned. *P. polysora* isolates were collected from different host plants. Since most of these host plants are in the association panel including 500 inbred lines, they were genotyped by molecular markers specific to *RppC* (Deng et al, 2022, Molecular Plant) and *RppK* (Table S2).

- (1) Hainan isolates were isolated from Dan340, KN5585, B73, Mo17, LX9801 and JK968 which are highly susceptible to SCR (Table S2). They do not carry *RppC* (Deng et al, 2022, Molecular Plant) and *RppK* gene (Table S2).
- (2) Guangxi isolates were isolated from local hybrids which were highly susceptible to SCR. Leaves containing a lot of *P. polysora* uredia were collected by local researchers and my students isolated spores from these leaf samples. But we do not know the names of these hybrids and did not check whether these local hybrids carry *RppC* or *RppK*.
- (3) Hubei isolates were isolated from maize inbred lines BY815, Zheng58, Liao138, 04K5686, YAN414 and JI53 which are highly susceptible to SCR (Table S2). They do not carry *RppC* (Deng et al, 2022, Molecular Plant) and *RppK* gene (Table S2).
- (4) *PP.CN1.0*, *PP.CN2.0*, and *PP.CN3.0* were isolated by Dr. Ding group at Henan Agricultural University and their information was mentioned in Deng et al, 2022, Molecular Plant. *PP.CN1.0*, *PP.CN2.0* and *PP.CN3.0* were isolated from three different unknown maize materials. We do not know their genotype information of *RppC* and *RppK*.
- (5) *PP.Hainan* was isolated from maize inbred line Dan340. DAN340 does not carry *RppC* (Deng et al, 2022, Molecular Plant) and *RppK* gene (Table S2).
- (6) *PP.Guangdong* was isolated from a local hybrid which were highly susceptible to SCR (SCR score >7). Leaves containing a lot of *P. polysora* uredia were collected by local researchers and my students isolated spores from these leaf samples. But we do not know the names of the hybrid and did not check whether the local hybrid carries *RppC* or *RppK*.
- (7) *PP.Wuhan* was isolated from maize inbred line BY815. BY815 does not carry *RppC* (Deng et al, 2022, Molecular Plant) and *RppK* gene (Table S2).

Reference: Deng et al., The *RppC*-*AvrRppC* NLR-effector interaction mediates the resistance to

3. Do you ever see any SCR on maize line K22? If so do those isolates carry AvrRppK?

Answer: Thank you for your comments. We did not see any SCR on maize line K22 during the last ten years. For each field test and inoculation in growth room, K22 was taken as the resistant control. And no SCR was observed on maize line K22.

4. We think that it is premature to conclude that AvrRppk is a virulence factor just based on the result that AvrRppK suppresses INF1-induced cell death in *N. benthamiana*. This is certainly an interesting piece of evidence, but it is, in the end, evidence that it suppresses the effects of a response in a foreign host to a protein that does not occur in the maize/*P. polysora* system- so it is interesting but not directly relevant.

Answer: Thank you very much for your comments.

In order to check whether AvrRppK is a virulent effector, we generated transgenic maize plants overexpressing *AvrRppK* ΔSP .

After inoculation with *P. polysora*, transgenic positive plants in two independent lines showed more susceptible phenotype to SCR than transgenic negative plants. It indicates that AvrRppK can enhance plant susceptibility to SCR (Fig. 4D-4F).

Also, we checked whether AvrRppK can suppress PTI by using these transgenic plants. After chitin treatment, strong PTI responses (MAP kinase activity and ROS accumulation) were activated in transgenic negative plants. While, MAP kinase activity and ROS accumulation triggered in transgenic plants overexpressing *AvrRppK* ΔSP were much weaker than these in transgenic negative plants (Fig. 4G-4H and Fig. S25).

So, we concluded that AvrRppK is a virulent factor and its expression in maize can suppress chitin-triggered PTI responses.

5. The authors should test a *P. polysora* knock out mutant which does not express AvrRppK in maize lines if they want to verify that AvrRppK is a virulence factor. In the absence of this evidence, they should be a lot more circumspect in their interpretation of their results.

Answer: Thank you for your comments. *P. polysora* is a biotroph and cannot be cultured on medium or genetically modified. So, we cannot generate *P. polysora* knockout mutants.

In order to check whether AvrRppK is a virulent effector, we generated transgenic maize plants overexpressing *AvrRppK* ΔSP .

After inoculation with *P. polysora*, transgenic positive plants in two independent lines showed more susceptible phenotype to SCR than transgenic negative plants. It indicates that AvrRppK can enhance plant susceptibility to SCR (Fig. 4D-4F).

Also, we checked whether AvrRppK can suppress PTI by using these transgenic plants. After chitin treatment, strong PTI responses (MAP kinase activity and ROS accumulation) were activated in transgenic negative plants. While MAP kinase activity and ROS accumulation triggered in transgenic plants overexpressing *AvrRppK* ΔSP were much weaker than these in transgenic negative plants (Fig. 4G-4H and Fig. S25).

So, we concluded that AvrRppK is a virulent factor and its expression in maize can suppress chitin-triggered PTI responses.

6. The definition of a “core effector” that the authors use is vague but it comes down to 2 things:

1. The effector is widespread in pathogen isolates

2. The effector is a virulence factor (a criterion which we think would apply to most effectors)

Not having any evidence to the contrary, we have to assume that *AvrRppk* is not under the selective pressure in the absence of *RppK* being deployed in the field. The fact that *AvrRppK* is widely found is very likely caused by a combination of the rarity of *RppK* among maize germplasm and possibly the way the authors conducted their sampling. If *RppK* were deployed in a widespread manner, do the authors think that the frequency of *AvrRppK* would decrease? If so, then we would argue it is just a regular effector, not a “core effector”. So we believe that the authors should be more careful in their claims about *RppK* being a core effector sensor and its ability to confer broad spectrum resistance. The fact is that, as with any R-gene, until it is deployed on a wide scale we don't really know how broad spectrum or durable it will be.

Answer: Thank you very much for your comments.

From 2011 to 2020, we planted K22 in different areas in China and K22 showed highly resistant phenotype to SCR (Fig. S1A). Also, we got five *P. polysora* isolates: *PP.CN1.0*, *PP.CN2.0*, *PP.CN3.0*, *PP.Guangdong* and *PP.Hainan*. After inoculation with these five isolates, *RppK* transgenic positive plants in two independent lines were highly resistant to them; while, transgenic negative plants were highly susceptible to them (Fig. S9). This indicates that *RppK*-mediated resistance does confer broad resistance against SCR. I agree with you that we do not know how broad spectrum or durable the *RppK*-mediated resistance will be until *RppK* is deployed on a wide scale.

Based on the references (Dangl et al. 2013, Science; Chepsergon et al. 2021. Virulence), core effector is based on two things (wide distribution across the population of a particular pathogen and its virulence function). So, core effectors should be widely distributed in many isolates of a particular pathogen. And it does not mean core effectors should be distributed in all isolates of a particular pathogen.

In order to check whether *AvrRppK* is widely distributed in *P. polysora* population, we collected more than 100 *P. polysora* isolates from different areas in Hainan province, Guangxi province and Hubei province. *AvrRppK* gene sequences with the same size were amplified from these isolates. Further, we sequenced 20 isolates from Hainan province, 11 isolates from Guangxi province and 6 isolates from Hubei province. Their *AvrRppK* gene sequences were identical to its in Wuhan isolate (Fig. 4A). In case of sampling bias, we mixed all spores isolated from Hainan province as Hainan-mix-isolates, mixed all spores isolated from Guangxi as Guangxi-mix-isolates, and mixed all spores isolated from Hubei province as Hubei-mix-isolates. DNA samples extracted from Hainan-mix-isolates, Guangxi-mix-isolates and Hubei-mix were used to sequence *AvrRppK* gene and the results showed that all sequences of *AvrRppK* were identical to it in Wuhan isolate (Fig. 4A). So, *AvrRppK* is broadly distributed in *P. polysora* population, but it does not mean *AvrRppK* gene exists in all isolates.

In order to check whether *AvrRppK* is a virulent effector, we generated transgenic maize plants overexpressing *AvrRppK* ΔSP . After inoculation with *P. polysora*, transgenic positive plants in two independent lines showed more susceptible phenotype to SCR than transgenic negative plants (Fig.

4D-4F). It indicates that *AvrRppK* can enhance plant susceptibility to SCR. Also, we checked whether *AvrRppK* can suppress PTI by using these transgenic plants. After chitin treatment, strong PTI responses (MAP kinase activity and ROS accumulation) were activated in transgenic negative plants. While MAP kinase activity and ROS accumulation triggered in transgenic plants overexpressing *AvrRppK* Δ *SP* were much weaker than these in transgenic negative plants (Fig.4G-4H and Fig. S25). So, we concluded that *AvrRppK* is a virulent factor and its expression in maize can suppress chitin-triggered PTI responses.

Taken together, we concluded that *AvrRppK* is a core effector of *P. polysora*.

References:

- (1) Dangl JL, Horvath DM, Staskawicz BJ. Pivoting the plant immune system from dissection to deployment. *Science*. 2013;341(6147):746-51. Doi: 10.1126/science.1236011.
- (2) Chepsergon J, Motaung TE, Moleleki LN. "Core" RxLR effectors in phytopathogenic oomycetes: A promising way to breeding for durable resistance in plants? *Virulence*. 2021;12(1):1921-1935. doi: 10.1080/21505594.2021.1948277.

7. We don't think that the title, "Deployment of a core effector sensor, *RppK*, in maize confers broad spectrum resistance against southern corn rust", gives the reader a good idea of what was actually done- The paper describes identification of both a resistance gene (not a "core effector sensor") and the corresponding *Avr* gene. We think the title should say this!

Answer: Thank you for your comments.

The title has been changed into "The *RppK-AvrRppK* interaction mediates maize broad resistance against southern corn rust"

Minor comments:

1) In line 125, you have to add 4H1028 in brackets.

Answer: Thank you for your comments. It has been corrected.

The sentence has been changed into "Among the 11 recombinant lines, five lines (4H1074, 4H1505, 4H1083, **4H1028** and 4H1213) carrying the *R3* gene showed resistance to SCR."

2) In line 131, R2 is 12.6 kb, not 11.6.

Answer: Thank you for your comment. Sorry about that mistake.

Yes, you are correct. The size of R2 genomic DNA sequence is 12.6 kb. It has been corrected.

3) In line 134, KN5585 is a susceptible maize line? If so, you need to give the simple information about it.

Answer: Thank you for your comments. Yes, KN5585 is susceptible to SCR. We added this information in line 138.

We changed the sentence into "The two fragments were then transformed into the maize inbred line KN5585, which is susceptible to SCR".

4) In line 163, 'These SCR-resistant JK968 lines are currently commercially cultivated in China.' should be deleted. You mentioned it in line 157 already.

Answer: Thank you for your comments. I am sorry. I did not write it clearly. And that sentence has

been deleted.

Maize hybrid JK968 (Jing724 × Jing92) has been widely planted on over seven million hectares in China over the past decade; and JK968 does not carry *RppK* gene and is susceptible to SCR (line 157).

We introgressed *RppK* gene from K22 into multiple hybrid lines and Some of these SCR-resistant lines are currently commercially cultivated in China. But those companies refused to give us any official verification, because it is a trade secret for any company who used *RppK* gene for breeding.

5) In line 799, JK968b should be added next JK968a. Fig2 (E) has the data with JK968b.

Answer: Thank you very much for your comments. According to your suggestion, we reorganized the data.

JK968b was added next JK968a. The data of JK968 were added in Fig 2E and Fig 2F; and its data were showed as a column with diagonal stripes.

6) ‘JK968b was derived from a cross between Jing724RppK and Jing92RppK (Fig. S10)’ should move to (E) legend.

Answer: Thank you for your comments.

The data of JK968 in Fig. S10 were moved into Fig 2E and Fig 2F; and its data were shown as a column with diagonal stripes.

7) In figure 3C, there is no positive control in *N benthamiana*. You can infiltrate *Rp1-D21* which you used in luciferase assay.

Answer: Thank you for your comments.

We repeated this experiment again. In this repeat, we infiltrated *Rp1-D21* as the positive control. Please check Figure 3C.

8) In supplementary data, what is 1145 in fig S3? Why did you use this line?

Answer: Thank you for your comments. Sorry about that, we deleted 1145 from Fig S3.

Maize inbred line 1145 is resistant to SCR. Before we got the BAC library of K22, we tried to identify candidate genes of *RppK* by screening the BAC library of 1145. But we did not use it after we identified the positive BAC clones from the BAC library of K22.

9) In Fig S8, (B) has four gel pictures. In Del13K marker, is it correct to write 13K, not DR3? Please confirm this.

Answer: Thank you for your comments.

Marker Del13K was used for detect DR3 gene. We corrected it and 13K was replaced by DR3.

Reviewer#3

This study describes the cloning of a disease resistance gene (designated RppK) and that of its corresponding avirulence gene (AvrRppK) in the maize-Southern corn rust (SCR) pathosystem. SCR is a serious disease of maize that seems to be getting worse and worse perhaps because of climate change or agronomic practices. Many maize genes and QTL

conferring resistance to SCR have been defined by genetic studies but none has been cloned thus far. So, this represents the first report of the successful cloning of an R gene that confers race-specific resistance to SCR.

RppK was found in a Chinese inbred K22, in which the authors claim the gene has been successful in conferring resistance over the past 30 years. On the basis of this observation they claim it to be a durable gene. A map based cloning approach was used to clone a candidate gene for RppK, which was subsequently validated by transgenesis to be the correct one. In addition, they showed that another inbred line SCML205 that was previously characterized to contain the RppS resistant locus actually contains an allele of RppK in that their genetic sequence is identical except for an intronic 2 bp indel in RppS. No surprise that the gene encodes a typical NLR, a CNL.

RppK was transferred to a number of susceptible elite inbreds, and the hybrids generated from them were shown to have superior yields in the presence of the disease and no yield penalty in its absence.

After having cloned RppK, the authors sought to clone the SCR effector gene that RppK intercepts to confer HR. The authors used an elegant approach to accomplish it. In brief, they first sequenced the genome of the rust pathogen (*Puccinia polysora*; PP, an isolate from Wuhan) and bioinformatically identified 965 genes predicted to encode host secreted proteins. Three hundred and thirty-eight of these genes were cloned and tested in a transient protoplast assay system to detect cell death (HR). Only one gene – designated PPG1259 – triggered a robust HR and was considered a candidate gene for AvrRppK. They next used the *Nicotiana benthamiana* heterologous system to show that co-infiltration of RppK and PPG1259 does indeed trigger an HR. Another assay that they used to provide further support to the correct cloning of the Avr gene, was based on the injection of purified proteins of PPG1259 and that of a putative effector gene PPG348 as a control in plants containing and lacking transgenic RppK. The observation that only PPG1259 caused cell death and that too in the plant containing the RppK transgene was deemed as confirmation that PPG1259 is AvrRppK. It encodes a 96-aa protein that exhibits no sequence identity with any known proteins or domains. Next, they showed that the genotype of the Avr1259 gene in more than 100 isolates of SCR that they collected from three different provinces (Hainan, Guansi and Hubei) was the same. The gene was then amplified and sequenced from 37 isolates (from all three provinces) and found to have the identical sequence. This high conservation of sequence was interpreted to mean that this gene represents a core effector and thus resistance against it is expected to be durable. Overall it is a decent manuscript, and I am convinced that the authors have definitely cloned the gene underlying RppK and perhaps also its corresponding avirulence gene, AvrRppk. However, there are a few concerns that I would like to bring up here.

1. First, I am not sure if the evidence is there yet to suggest that AvrRppk is a core effector. Sure, they did show that there is a great deal of sequence conservation, but they provided no evidence of the race structure of these isolates. One possibility is that there is very little diversity in SCR in China. It would have been better if they also sequenced a couple of other effector genes in addition to PPG1259 to address their conservation.

Answer: Thank you for your comments.

Up to now, only one effector gene (*AvrRppC*) was reported in *P. polysora* (Deng et al. 2022,

Molecular Plant). We sequenced the effector gene *AvrRppC* in 7 *P. polysora* isolates and identified seven different *AvrRppC* alleles (*AvrRppC^{ref}*, *AvrRppC^A*, *AvrRppC^C*, *AvrRppC^E*, *AvrRppC^F*, *AvrRppC^J* and *AvrRppC^L*) (Fig9A). It indicates that there are a lot of diversity in *P. polysora* isolates in China. The results were consistent with the data reported before (Deng et al. 2022, Molecular Plant). Also, we inoculated transgenic plants containing *RppK* gene with five *P. polysora* isolates. The Results showed that transgenic positive plants were highly resistant to these five isolates; while, transgenic negative plants were highly susceptible to these isolates.

Reference: Deng et al., The RppC-AvrRppC NLR-effector interaction mediates the resistance to southern corn rust in maize, Molecular Plant (2022), <https://doi.org/10.1016/j.molp.2022.01.007>

2. The experiment they did to show that AvrRppK has virulence activity is also concerning. They used suppression of cell death mediated by INF1, which they say is a typical PAMP, as an assay for the virulence activity of AvrRppK. I don't think induction of cell death is a typical PTI response. Regardless, it would have been more convincing if they also looked at the effect of AvrRppK on Rp1-D21-mediated cell death.

Answer: Thank you for your comments. We added more evidences to support that AvrRppK is a virulence factor.

In order check whether AvrRppK is a virulent effector, we generated transgenic maize plant overexpressing *AvrRppK ΔSP*.

After inoculation with *P. polysora*, transgenic positive plants in two independent lines showed more susceptible phenotype to SCR than transgenic negative plants. It indicates that AvrRppK can enhance plant susceptibility to SCR (Fig. 4D-4F).

Also, we checked whether AvrRppk can suppress PTI by using these transgenic plants. After chitin treatment, strong PTI responses (MAP kinase activity and ROS accumulation) were activated in transgenic negative plants. While, MAP kinase activity and ROS accumulation triggered by chitin in transgenic plants overexpressing *AvrRppK ΔSP* were much weaker than these in transgenic negative plants (Fig.4G-4H and Fig. S25).So, we concluded that AvrRppK is a virulent factor and its expression in maize can suppress chitin-triggered PTI responses.

Also, we checked the effect of AvrRppK on *Rp1-D21*-mediated cell death. And the results showed that transient expression of *AvrRppK* in *N. benthamiana* cannot suppress *Rp1-D21*-mediated cell death (Fig.S24).

3. I could not understand the rationale for doing the single-cell sequencing technology experiment, what they got out of it, and why it matters for this work.

Answer: Thank you very much for your comments. In this case, we did not try to emphasize the single-cell technology. We just got the genomic DNA sequence of *AvrRppK* gene in *P. polysora* by using the single-cell sequencing technology.

At the beginning, we did not know it is conserved in all tested isolates. What we planned to do at that time was to clone its genomic DNA sequence. However, we failed to get it after tried multiple ways. Since single-cell sequencing technology is an option to get it, we tried and we finally got the genomic DNA sequence of *AvrRppK*. So, the single-cell sequencing technology was used only for getting the genomic DNA sequence of *AvrRppK*.

4. I would have appreciated if the authors discussed the similarities, if any, in the ancestry of the 17 lines of the diversity panel. Are any of these lines from places other than China? How prevalent was the *RppK* gene in the commercial germplasm? I think these questions are relevant given the claim that this R gene has been durable for more than 30 years.

Answer: Thank you very much for your comments.

The *RppK* genes in the sixteen lines showed high identity with that in K22 and only five SNPs were identified among the *RppK* genes of seventeen lines. In the previous report (Yang et al., 2011, Molecular breeding), a neighbor-joining phylogenetic tree generated based on Nei's genetic distance displayed the relationship among 500 inbred lines. Based on the neighbor-joining phylogenetic tree, the 500 inbred lines were classified into four subgroups: Stiff stalk (SS), non-stiff stalk (NSS), tropical-subtropical (TST) and mixed groups. According to the results in that reference (Yang et al., 2011, Molecular breeding), six of the seventeen lines (K22, 526018, LXN, M97, 1462, and DAN9046) belong to NSS subgroup; eight of them (CIMBL134, CIMBL135, CML432, SW1611, YUN46, CIMBL35, CML433 and CIMBL63) belong to TST subgroup, three of them (975-12, CHENG698 and Z2018F) belong to mixed subgroup and none of them belong to SS subgroup. This indicates that there is very low level of similarities between the ancestries of the 17 lines of the diversity panel.

In the seventeen lines containing *RppK* gene, eleven of them (K22, 526018, 975-12, CHENG698, LXN, M97, SW1611, YUN46, Z2018F, 1462 and DAN9046) are from China. Six of them (CIMBL134, CIMBL135, CML432, CIMBL35, CML433 and CIMBL63) are from CIMMYT, Mexico (Table S9).

We also checked 74 commercial maize hybrids in China by using molecular marker R8.63 and we found that only five of them (LiaoDan707, KangNongYu598, MeiGu555, Gaoyu14022 and KangNong2) contain *RppK* gene (Fig. S11 and Table S6). Based on the information on <https://chinaseed114.com/>, the five hybrids (LiaoDan707, KangNongYu598, MeiGu555, Gaoyu14022 and KangNong2) were released to the market in 2020, 2017, 2018, 2017 and 2010, respectively. This means *RppK* genes has not been broadly used in maize breeding. These might partially explain why *RppK* gene has been durable for more than 30 years.

We added those into the discussion section (Line352-374).

Reference: Yang, X., Gao, S., Xu, S. *et al.* Characterization of a global germplasm collection and its potential utilization for analysis of complex quantitative traits in maize. *Mol Breeding* 28, 511–526 (2011). <https://doi.org/10.1007/s11032-010-9500-7>

Reviewers' Comments:

Reviewer #1:

Remarks to the Author:

The manuscript by Chen et al., reports the identification of a novel RppK of CNL resistance protein from the maize inbred line K22 and its cognate avirulent effector AvrRppK from *P. polysora*. They provide significant amount of evidences to try to support the conclusion that a pair of a core effector and its cognate NLR interaction mediates maize broad resistance against southern corn rust. Two major concerns include:

1. They did a great deal of screening for AvrRppK, and co-expression or co-infiltration of RppK and AvrRppK activate HR in *N. benthamiana* or maize protoplast. While it still not clear whether the relationship between RppK and AvrRppK mediates maize resistance without genetic evidences of the pathogen.
2. As so far, evidences are still not enough to support the conclusions that "AvrRppk is a core effector" and "AvrRppK is a virulence effector".

Reviewer #2:

Remarks to the Author:

We think the authors did a good job responding to our comments. We just have a few minor criticisms/requests:

It was not specified which susceptible line was used for the transgenic line overexpressing AvrRppK in the main text.

There is no information on how the transgenic line overexpressing AvrRppK was developed in materials and methods.

The method used for the western blot to detect MAPK and AvrRppK is not described in materials.

How was the chitin treatment performed in the transgenic maize plant overexpressing AvrRppK. This should be noted in the materials and methods.

Reviewer #3:

Remarks to the Author:

After having read the revised manuscript and the authors' response to reviewer concerns, I feel the authors have done a commendable job in their efforts to address most concerns that I raised at least. Given this, I don't see any reason for not recommending it for publication.

Response to comments

Comments from reviewers

Reviewer #1 (Remarks to the Author):

The manuscript by Chen et al., reports the identification of a novel RppK of CNL resistance protein from the maize inbred line K22 and its cognate avirulent effector AvrRppK from *P. polysora*. They provide significant amount of evidences to try to support the conclusion that a pair of a core effector and its cognate NLR interaction mediates maize broad resistance against southern corn rust. Two major concerns include:

1. They did a great deal of screening for AvrRppK, and co-expression or co-infiltration of RppK and AvrRppK activate HR in *N. benthamiana* or maize protoplast. While it still not clear whether the relationship between RppK and AvrRppK mediates maize resistance without genetic evidences of the pathogen.

Answer: Thank you for your comments.

We deleted the conclusion about the relationship of *RppK* and *AvrRppK* in activating resistance response.

We changed the title into “*RppK* Mediates Maize Resistance against Southern Corn Rust through Its Cognate Gene *AvrRppK*”.

2. As so far, evidences are still not enough to support the conclusions that “AvrRppk is a core effector” and “AvrRppK is a virulence effector”.

Answer: Thank you for your comments.

We deleted the conclusion about “AvrRppK is a core effector” and “AvrRppK is a virulence effector”.

Reviewer #2 (Remarks to the Author):

We think the authors did a good job responding to our comments. We just have a few minor criticisms/requests:

It was not specified which susceptible line was used for the transgenic line overexpressing AvrRppK in the main text.

Answer: Thank you very much for your comments.

We transferred the *AvrRppK* ΔSP overexpression construct into the maize inbred line KN5585. And this information has been added in the materials and methods.

There is no information on how the transgenic line overexpressing AvrRppK was developed in materials and methods.

Answer: Thank you very much for your comments.

We added the information about developing *AvrRppK* ΔSP overexpression transgenic plants in the materials and methods.

The method used for the western blot to detect MAPK and AvrRppK is not described in materials.

Answer: Thank you very much for your comments.

We added the method used for the western blot to detect MAPK and AvrRppK in the materials and

methods.

How was the chitin treatment performed in the transgenic maize plant overexpressing AvrRppK. This should be noted in the materials and methods.

Answer: Thank you very much for your comments.

We added the information about the chitin treatment in the materials and methods.

Reviewer #3 (Remarks to the Author):

After having read the revised manuscript and the authors' response to reviewer concerns, I feel the authors have done a commendable job in their efforts to address most concerns that I raised at least. Given this, I don't see any reason for not recommending it for publication.

Answer: Thank you very much for your comments and your time.